# $\varepsilon$-Optimally Solving Two-Player Zero-Sum POSGs

**Erwan C. Escudie**
e.c.escudie@rug.nl
University of Groningen

**Matthia Sabatelli**
m.sabatelli@rug.nl
University of Groningen

**Olivier Buffet**
olivier.buffet@loria.fr
Inria – Nancy Grand-Est

**Jilles Steeve Dibangoye**
j.s.dibangoye@rug.nl
University of Groningen

## Abstract

We present a novel framework for $\varepsilon$-optimally solving two-player zero-sum partially observable stochastic games (zs-POSGs). These games pose a major challenge due to the absence of a principled connection with dynamic programming (DP) techniques developed for two-player zero-sum stochastic games (zs-SGs). Prior attempts at transferring solution methods have lacked a lossless reduction—defined here as a transformation that preserves value functions, equilibrium strategies, and optimality structure—thereby limiting generalisation to ad hoc algorithms. This work introduces the first lossless reduction from zs-POSGs to transition-independent zs-SGs, enabling the principled application of a broad class of DP-based methods. We show empirically that point-based value iteration (PBVI) algorithms, applied via this reduction, produce $\varepsilon$-optimal strategies across a range of benchmark domains, consistently matching or outperforming existing state-of-the-art methods. Our results open a systematic pathway for algorithmic and theoretical transfer from SGs to partially observable settings.

## 1 Introduction

Bellman [1957] introduced the principle of optimality for sequential decision-making under uncertainty in the 1950s, originally in the context of Markov decision processes (MDPs). Since then, this principle has provided a foundation for solving progressively more complex problems. Shapley [1953] extended it to zero-sum stochastic games (zs-SGs), while others adapted it to the partially observable case, including Aström [1965], Smallwood and Sondik [1973], and Sondik [1978] for partially observable Markov decision processes (POMDPs). More recently, a rich body of work has applied this reduction-based methodology to partially observable stochastic games (POSGs). For common-payoff POSGs, several approaches have successfully constructed fully observable surrogates—typically common-payoff Markov games—thus enabling the transfer of dynamic programming (DP) theories and algorithms without compromising optimality [Szer et al., 2005, Oliehoek et al., 2008, 2010, Nayyar et al., 2013, Dibangoye et al., 2013a, 2016, Oliehoek, 2013, Oliehoek et al., 2013, Lerer et al., 2020, Peralez et al., 2024, 2025]. In contrast, for zs-POSGs, although a number of methods have been proposed [Wiggers et al., 2016, Nayyar and Gupta, 2017, Horák et al., 2017, Horák and Bošanský, 2019, Buffet et al., 2020, Brown et al., 2020, Delage et al., 2023, Sokota et al., 2023], none constitutes a lossless reduction [Sanjari et al., 2023]. As a consequence, generalisation to this setting has remained restricted to ad hoc algorithmic designs, with no principled framework for transferring DP techniques from SGs to zs-POSGs.

A lossless reduction transforms zs-POSGs into zs-SGs while satisfying three main criteria: value preservation, equilibrium correspondence, and information structure equivalence [Sanjari et al., 2023]. Value preservation requires that the expected return of any joint policy in the original game equals

that of its image in the reduced game. Equilibrium correspondence demands that the reduction induce a bijection between equilibria, ensuring that Nash strategies remain valid and interpretable across both formulations. Finally, information structure equivalence ensures that the transformation does not introduce extraneous information or collapse distinctions essential to the players' strategic reasoning. Several reductions have been proposed, but all fail to satisfy one or more of these criteria. The occupancy Markov game (OMG) assumes a centralised planner selects joint policies based on the occupancy state—an object unobservable to either player—thus violating the equilibrium correspondence criterion [Wiggers et al., 2016, Buffet et al., 2020, Delage et al., 2023]. To circumvent this limitation, Delage et al. [2023] introduce policy tracking via ad hoc bookkeeping techniques. The public-belief alternating Markov game (PuB-AMG) assumes that both players publicly commit to their policies, which violates the equilibrium correspondence criterion [Sokota et al., 2023]. To address this, the authors introduce a regularised minimax formulation intended to restore solution correspondence by ensuring that strategies computed in the reduced game are interpretable in the original zs-POSG. While the regularisation guarantees convergence to unique solutions within the PuB-AMG, the resulting strategies may correspond to policies with high exploitability in the original game. Although an annealing scheme can reduce this gap empirically, there is no formal guarantee that the regularised solutions preserve value or recover exact Nash equilibria in the original zs-POSG. As a result, the reduction does not satisfy the criteria for a lossless transformation.

**Contributions.** We make several key contributions to the study of zs-POSGs:

**(1) A principled and lossless reduction to transition-independent zs-SGs.** We introduce the first reduction that maps any zs-POSG to a strategically equivalent *transition-independent* zs-SG, preserving value, equilibrium structure, and information constraints. The reduction adopts a decentralised perspective: each player independently selects a sequence of decision rules—mappings from private histories (i.e., past actions and observations) to action distributions—defining the local state of that player, formalised as an *occupancy set*. The global state of the reduced game, the *occupancy state*, is the intersection of the two players' occupancy sets, capturing the joint consistency of their behaviours. Because each local state evolves independently of the opponent, the reduced game exhibits *transition-independent dynamics*, where transitions depend only on the current local state and selected decision rule. This reformulation preserves the strategic structure of the original zs-POSG while enabling dynamic programming over occupancy sets—avoiding explicit reasoning over joint policy spaces and supporting scalable, exact solution methods. The hierarchy of planners introduced in this work—ranging from focal to marginal planners—forms a nested structure based on increasing information availability, as illustrated in Figure 1.

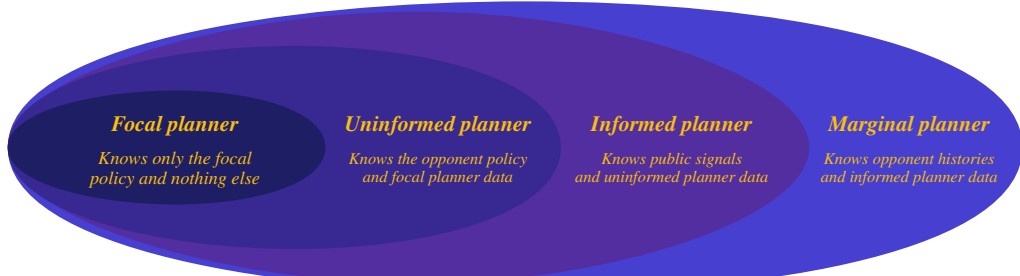

Figure 1: A planner hierarchy induced by relaxing information constraints, from the focal to the marginal planner, supporting our theoretical and algorithmic framework.

**(2) A planner hierarchy for structured reasoning.** The reduced game reveals a hierarchy of planners—ranging from a minimally informed *focal planner*, to increasingly informed *uninformed*, *informed*, and finally *marginal planners*. Each planner defines a distinct optimisation problem, characterised by its reasoning scope (single-agent or centralised) and its access to information (from no observations to full access to public and private histories). While only the focal planner is ever implemented in practice, the remaining planners serve as conceptual tools that underpin the structure of value functions and guide the transfer of theoretical insights. Solving the reduced game requires traversing this hierarchy: each planner contributes a well-defined subproblem whose value function and policy are essential for constructing the overall solution to the zs-POSG.

**(3) Structural properties of value functions.** The planner hierarchy reveals new structural properties of zs-POSGs, including *optimality equations*, *strategy selection rules*, and, critically, the *uniform continuity* of value functions. Uniform continuity guarantees that small changes in *occupancy states* lead to uniformly bounded changes in value, regardless of where they occur in the state space. This property enables value functions to generalise across occupancy states in a principled way, supporting reliable planning without requiring dense sampling or finely tuned control at every point.

**(4) Practical benefits through algorithmic transfer.** As a concrete example of the framework in use, we show that point-based value iteration (PBVI) [Pineau et al., 2003, Horák et al., 2017, Horák and Bošanský, 2019], applied to the reduced game, computes $\varepsilon$-optimal strategies across standard zs-POSG benchmarks, consistently matching or outperforming existing methods. More broadly, the reduction enables the transfer of a wide class of dynamic programming algorithms—originally developed for stochastic games—into partially observable settings, thereby expanding the set of scalable planning tools available for zs-POSGs.

## 2 Preliminaries

This section presents the standard formulation of zero-sum partially observable stochastic games (zs-POSGs), along with their associated policies and value functions.

**Definition 2.1.** *A two-player zero-sum partially observable stochastic game $\mathcal{M}$ is defined as the tuple $(\mathcal{S}, \mathcal{A}_1, \mathcal{A}_2, \mathcal{Z}_1, \mathcal{Z}_2, \mathcal{W}, p, r, b, \gamma, \ell)$, where players 1 and 2 are the maximising and minimising players, respectively. $\mathcal{S}$ is a finite set of hidden states. $\mathcal{A}_1$ and $\mathcal{A}_2$ are finite sets of private actions, and $\mathcal{Z}_1$ and $\mathcal{Z}_2$ are finite sets of private observations for each player. $\mathcal{W}$ denotes the set of public observations available to both players. The transition function $p : \mathcal{S} \times \mathcal{A}_1 \times \mathcal{A}_2 \to \Delta(\mathcal{S} \times \mathcal{Z}_1 \times \mathcal{Z}_2 \times \mathcal{W})$ defines the probability $p(s\prime, z_1, z_2, w | s, a_1, a_2)$ of transitioning to next state $s\prime$ and emitting observations $(z_1, z_2, w)$ given current state $s$ and actions $(a_1, a_2)$. The reward function $r : \mathcal{S} \times \mathcal{A}_1 \times \mathcal{A}_2 \to \mathbb{R}$ specifies the stage payoff $r(s, a_1, a_2)$ received by player 1. The initial belief over states is given by $b \in \Delta(\mathcal{S})$, the discount factor is $\gamma \in [0, 1)$, and the planning horizon is finite with $\ell + 1 < \infty$.*

**Policies.** At each stage $t \in \{0, \dots, \ell\}$, player $i$ selects actions based on a private action-observation history $h_{i,t} \in \mathcal{H}_{i,t} \doteq (\mathcal{A}_i \times \mathcal{Z}_i)^t$ and a public observation history $h_{\text{pub},t} \in \mathcal{H}_{\text{pub},t} \doteq \mathcal{W}^t$, starting from $h_{i,0} = \emptyset$. A decision rule $d_{i,t} : \mathcal{H}_{i,t} \times \mathcal{H}_{\text{pub},t} \to \Delta(\mathcal{A}_i)$ maps *joint* histories to distributions over actions, with the set of all such rules denoted $\mathcal{D}_{i,t}$. The players' actions determine a transition to state $s_{t+1}$, yield a payoff $r(s_t, a_{1,t}, a_{2,t})$, and generate new observations $(z_{1,t+1}, z_{2,t+1}, w_{t+1})$, which update the histories recursively. A policy $\pi_i = (d_{i,0}, \dots, d_{i,\ell})$ is a sequence of such rules; the set of all history-dependent policies is denoted $\Pi_i$. The full sets of private and public histories are $\mathcal{H}_i = \cup_{t=0}^{\ell} \mathcal{H}_{i,t}$ and $\mathcal{H}_{\text{pub}} = \cup_{t=0}^{\ell} \mathcal{H}_{\text{pub},t}$, respectively.

**Value Functions.** Given an initial state distribution $b$, the expected cumulative discounted payoff under joint policies $(\pi_1, \pi_2)$ is $v_{\pi_1, \pi_2}(b) = \mathbb{E}[\sum_{t=0}^{\ell} \gamma^t \cdot r(s_t, a_{1,t}, a_{2,t})]$, where the expectation is over trajectories induced by $b$, $p$, and the policy pair. Player 1 seeks to maximise this value while player 2 seeks to minimise it. Under perfect recall, behavioural (history-dependent) policies are equivalent to mixed strategies [Kuhn, 1953], and von Neumann's minimax theorem [Neumann, 1928]—extended to behavioral strategy spaces by Delage et al. [2023]—ensures the existence of a game value $v_*(b)$, satisfying $v_*(b) = \min_{\pi_2} \max_{\pi_1} v_{\pi_1, \pi_2}(b) = \max_{\pi_1} \min_{\pi_2} v_{\pi_1, \pi_2}(b)$. The solution to $\mathcal{M}$ is a policy $\pi_1$ that maximises the guaranteed payoff against any opponent policy, i.e., $\min_{\pi_2} v_{\pi_1, \pi_2}(b) = v_*(b)$; the symmetric holds for player 2. The corresponding pair forms a Nash equilibrium.

**Lossless Reductions.** A reduction from a zs-POSG $\mathcal{M}$ to a surrogate game $\mathcal{M}'$ is said to be *lossless* if it preserves value functions, supports equilibrium transfer, and maintains the relevant information structure. This includes: (i) *value preservation*, i.e., $v_{\pi_1, \pi_2}^{\mathcal{M}}(b) = v_{\pi_1, \pi_2}^{\mathcal{M}'}(b)$ for all joint policies; (ii) *equilibrium correspondence*, meaning each Nash equilibrium in $\mathcal{M}'$ induces one in $\mathcal{M}$ and vice versa; and (iii) *information compatibility*, ensuring the reduction respects players' original observation constraints and decision spaces. These conditions allow exact transfer of optimality equations and policies, while preserving the strategic essence of the original game.

# 3 Reducing zs-POSGs to Transition-Independent zs-SGs

This section presents the main contribution of this work: a lossless reduction from any zs-POSG to a strategically equivalent transition-independent zs-SG. The reduced model preserves the value function, equilibrium structure, and information constraints of the original zs-POSG, thereby enabling the exact transfer of dynamic programming principles and solution methods. The key insight behind this reduction is to factor the game into local planning processes, one for each player, while maintaining the strategic dependencies of the original interaction through a shared global state.

We reformulate the original zs-POSG as a planning process—a transition-independent zs-SG—through two complementary perspectives. The centralised view casts it as a planning problem executed by an *uninformed planner*, a hypothetical central authority that selects joint decision rules without access to any observations. The decentralised view decomposes this process into two *player-specific focal planners*, each reasoning independently over a single player's sequence of decision rules. This planning process unfolds stage by stage. At each time step $t$, the underlying global state is an *uninformed occupancy state* $x_t$: a distribution over hidden states, private action-observation histories, and public observations, induced by the sequence of decision rules $\theta_t \doteq (d_{1,0}, d_{2,0}, \ldots, d_{1,t-1}, d_{2,t-1})$. This global state captures the full strategic context of the game but remains unobservable to either player. Instead, each player reasons over a local state $x_{i,t}$, defined as a player-specific *occupancy set*: the collection of uninformed occupancy states consistent with their own sequence of decision rules $(d_{i,0}, \ldots, d_{i,t-1})$, regardless of the opponent's choices. Based solely on this local state, player $i$ selects a decision rule $d_{i,t}$ and transitions to the next local state $\tau_i(x_{i,t}, d_{i,t})$, formed by appending the selected rule. This decentralised process continues until the planning horizon $\ell + 1$ is reached. At each step, the environment returns an immediate payoff $\rho(x_t, d_{1,t}, d_{2,t})$ and updates the global state via $\tau(x_t, d_{1,t}, d_{2,t})$, both unobservable to the players. Crucially, each local state evolves independently of the opponent, and the current global state satisfies $\{x_t\} = x_{1,t} \cap x_{2,t}$. These properties define a structured model known as a *transition-independent zero-sum stochastic game (zs-SG)*, *cf.* Figure 2.

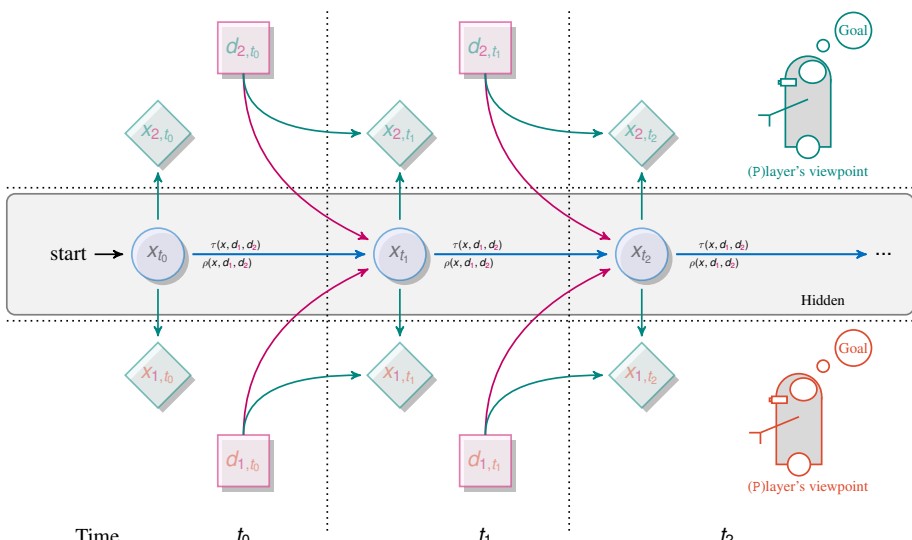

Figure 2: An influence diagram of a transition-independent, two-player, zero-sum stochastic game.

We first describe model $\mathcal{M}_i$ for each player-specific *focal planner*—a single-agent planner that selects policies for player $i$, based only on their own decision-rule history, with no access to observations or the opponent's policy. This defines the least informed level of the planner hierarchy and serves as the local computational engine underlying decentralised dynamic programming within the zs-SG.

**Definition 3.1.** *A player-specific focal planning process $\mathcal{M}_i = (\mathcal{X}_i, \mathcal{F}_i, \mathcal{D}_i, \tau_i, \rho_i)$ consists of: a set $\mathcal{X}_i$ of local states (occupancy sets); a set $\mathcal{F}_i \subset \mathcal{X}_i$ of (terminal) occupancy sets at stage $\ell + 1$; a set of decision rules $\mathcal{D}_i$; a local transition operator $\tau_i(x_i, d_i) = \{\tau(x, d_i, d_{-i}) | x \in x_i, d_{-i} \in \mathcal{D}_{-i}\}$, and a local payoff function, which is zero if $x_i \in \mathcal{X}_i \setminus \mathcal{F}_i$ and $\rho_i: x_i \mapsto \overline{\mathrm{opt}}_{x \in x_i} g(x)$ otherwise, where*

*the operator* $\overline{\text{opt}}$ *corresponds to* $\min$ *for player* 1 *and* $\max$ *for player* 2*, and, for any uninformed occupancy state* $x_{t+1}$*,* $g(x_{t+1}) \doteq \mathbb{E}_{(s_0,a_{1,0},a_{2,0},\ldots,s_t,a_{1,t},a_{2,t},s_{t+1})\sim\Pr(\cdot|x_{t+1})}[\sum_{t'=0}^{t} \gamma^{t'} \cdot r(s_{t'}, a_{1,t'}, a_{2,t'})]$.

Having formalised the focal planning processes as $\mathcal{M}_1$ and $\mathcal{M}_2$, we now lift these constructions to define the transition-independent zero-sum stochastic game $\mathcal{M}'$, which governs the joint dynamics over uninformed occupancy states and their associated objective.

**Definition 3.2.** *A transition-independent zero-sum stochastic game (zs-SG) is a tuple* $\mathcal{M}' = (\mathcal{X}, \mathcal{F}, \mathcal{M}_1, \mathcal{M}_2, \tau, \varphi, \rho, \ell, \gamma)$*, where* $\mathcal{X}$ *is the set of uninformed occupancy states;* $\mathcal{F} \subset \mathcal{X}$ *is the set of (terminal) uninformed occupancy states at stage* $\ell + 1$*;* $\tau : (x, d_1, d_2) \mapsto x\prime$ *is the global transition function, that is, for all hidden states* $s\prime$*, private action-observation histories* $(h_i, a_i, z_i)$ *for player* $i$*, and public observation histories* $(h_{pub}, w)$*,*

$$x\prime(s\prime, (h_i, a_i, z_i)_i, (h_{pub}, w)) = \sum_s x(s, h_1, h_2, h_{pub})p(s\prime, z_1, z_2, w|s, a_1, a_2)\prod_i d_i(a_i|h_i, h_{pub});$$

$\varphi(x_1, x_2) = x$*, where* $\{x\} = x_1 \cap x_2$*;* $\rho : (x, d_1, d_2) \mapsto \mathbb{R}$ *is the stage-wise payoff function, given by*

$$\rho(x, d_1, d_2) = \sum_s \sum_{h_1,h_2} \sum_{h_{pub}} x(s, h_1, h_2, h_{pub}) \sum_{a_1,a_2} d_1(a_1|h_1, h_{pub})d_2(a_2|h_2, h_{pub})r(s, a_1, a_2);$$

$\ell + 1$ *is the horizon and* $\gamma \in [0, 1)$ *the discount factor. Each component* $\mathcal{M}_i$ *captures the planning process from the perspective of player* $i$*, see Definition 3.1.*

Having defined the transition-independent zs-SG $\mathcal{M}'$, we now specify the objective of solving it. The goal is to find policies $\psi_1 : \mathcal{X}_1 \to \mathcal{D}_1$ and $\psi_2 : \mathcal{X}_2 \to \mathcal{D}_2$ that map player-specific occupancy sets to decision rules. These are *occupancy-set dependent* policies, tailored to the decentralised structure of the reduced game. Let $\Psi_i$ denote the set of such policies for player $i$. Given an initial uninformed occupancy state $x_0 = b$ and initial occupancy sets $x_{1,0} = x_{2,0} = \{b\}$, the expected cumulative discounted payoff under joint policy $(\psi_1, \psi_2)$ is defined as

$$v'_{\psi_1,\psi_2}(b) \doteq \sum_{t=0}^{\ell} \gamma^t \cdot \rho(x_t, d_{1,t}, d_{2,t})|x_t = \varphi(x_{1,t}, x_{2,t}), d_{i,t} = \psi_i(x_{i,t}), x_{i,t} = \tau_i(x_{i,t-1}, d_{i,t-1}).$$

Player 1 seeks to maximise this quantity, while player 2 aims to minimise it. We now show that the reduced game $\mathcal{M}'$ admits a well-defined value and satisfies the minimax property, enabling us to reason about optimal policies via standard game-theoretic principles.

**Lemma 1.** *The reduced game* $\mathcal{M}'$ *admits a well-defined value* $v'_*(b)$*, which satisfies the minimax identity:* $v'_*(b) = \min_{\psi_2 \in \Psi_2} \max_{\psi_1 \in \Psi_1} v'_{\psi_1,\psi_2}(b) = \max_{\psi_1 \in \Psi_1} \min_{\psi_2 \in \Psi_2} v'_{\psi_1,\psi_2}(b).$

The objective in solving $\mathcal{M}'$ is to compute an optimal policy $\psi_i^*$ for each player $i$ such that

$$\min_{\psi_2 \in \Psi_2} v'_{\psi_1^*,\psi_2}(b) = \max_{\psi_1 \in \Psi_1} v'_{\psi_1,\psi_2^*}(b) = v'_*(b).$$

We now formally state our main theoretical result, which establishes that the reduced game satisfies the lossless reduction criteria introduced above.

**Theorem 1.** *The reduced game* $\mathcal{M}'$ *constitutes a lossless reduction of the original zs-POSG* $\mathcal{M}$*.*

This reformulation preserves the strategic structure of the original zs-POSG while enabling dynamic programming across a hierarchy of planners, avoiding explicit reasoning over joint policy spaces and supporting scalable, $\varepsilon$-exact solution methods.

# 4 Solving Transition-Independent zs-SGs via A Hierarchy of Planners

Transition-independent zs-SGs enable value-based planning over structured state spaces induced by sequences of decision rules selected independently by each player. This structure supports a *hierarchy of planners*, from local focal planners reasoning unilaterally over a single player's policy to more informed marginal and central planners. While the two least-informed planners suffice to define the reduced game, the full hierarchy enables more efficient solutions. Foundational results in planning and reinforcement learning show that hierarchical formulations improve efficiency by introducing abstraction, decomposition, and temporally extended reasoning [Ghallab et al., 2004, André and Russell, 2002, Vezhnevets et al., 2017, Kaelbling and Lozano-Pérez, 2011]. Similarly, in transition-independent zs-SGs, the planner hierarchy refines state representations—from player-specific occupancy sets to marginal occupancy states—while preserving strategic structure and

enabling dynamic programming. The hierarchy includes two *focal planners*, one per player; a central *uninformed planner* blind to observations; an *informed planner* with access to public signals; and two *marginal planners*, each aware of both policies but only one player's private trajectory. This layered structure underpins the analysis and algorithms presented next.

Solving a transition-independent zs-SG can be approached by solving focal planning problems, each defined over a player-specific process $\mathcal{M}_i$. These problems may be solved independently—when computing only a safe policy for one player—or jointly, since the planners share structural components. The objective is to compute an optimal occupancy-set dependent policy $\psi_i^*$ that optimises the worst-case expected return. The value of a given policy $\psi_i$ from initial state $x_{i,0}$ is $v_{i,\psi_i}(x_{i,0}) = \rho_i(x_{i,\ell+1})$, where $x_{i,t} = \tau_i(x_{i,t-1}, \psi_i(x_{i,t-1}))$. The optimal value function is then $v_{i,*}(x_{i,0}) = \mathrm{opt}_{\psi_i \in \Psi_i} v_{i,\psi_i}(x_{i,0})$. The following result characterises this function through [Bellman](#)'s optimality equations.

**Theorem 2.** *The optimal state-value function* $v_{i,*}: \mathcal{X}_i \to \mathbb{R}$ *of* $\mathcal{M}_i$ *satisfies [Bellman](#)'s optimality equations:* $v_{i,*}(x_i) = \rho_i(x_i)$ *if* $x_i \in \mathcal{F}_i$*, and* $v_{i,*}(x_i) = \mathrm{opt}_{d_i \in \mathcal{D}_i} v_{i,*}(\tau_i(x_i, d_i))$ *otherwise; with an optimal policy given by* $\psi_i^*: x_i \mapsto \arg\mathrm{opt}_{d_i \in \mathcal{D}_i} v_{i,*}(\tau_i(x_i, d_i))$*, where the optimisation operator* $\mathrm{opt}$ *corresponds to* $\max$ *for player* 1 *and* $\min$ *for player* 2.

Focal planners offer safe and implementable policies but are difficult to solve due to reasoning over entire occupancy sets. We now turn to the next planner in the hierarchy: the uninformed planner. This planner operates over individual uninformed occupancy states and, while not designed to extract a safe policy for either player, its value function aligns with that of the focal planners. Specifically, $v_{1,*}(x_{1,0}) = v_{2,*}(x_{2,0}) = v_*(x_0)$, where $x_0 = \varphi(x_{1,0}, x_{2,0})$. Thus, computing the value of an occupancy set can be reduced to computing the values of its constituent uninformed occupancy states.

**Theorem 3.** *The optimal state-value function* $v_*: \mathcal{X} \to \mathbb{R}$ *of* $\mathcal{M}'$ *satisfies [Bellman](#)'s optimality equations:* $v_*(x) = 0$ *if* $x \in \mathcal{F}$*, and* $v_*(x) = \max_{d_1 \in \mathcal{D}_1} \min_{d_2 \in \mathcal{D}_2} [\rho(x, d_1, d_2) + \gamma v_*(\tau(x, d_1, d_2))]$ *otherwise. For player* $i$*, the value of their focal planner at occupancy set* $x_i$ *at stage* $t$ *is given by:*

$$v_{i,*}(x_i) = \overline{\mathrm{opt}}_{x \in x_i} [g(x) + \gamma^t v_*(x)], \quad \forall x_i \in \mathcal{X}_i.$$

The uninformed planner treats all public observation histories as indistinguishable, preventing it from leveraging structure revealed by public signals. The *informed planner* addresses this limitation by reasoning separately for each realisation of public observations. It operates over *informed occupancy states* $o_{x,h_{\mathrm{pub}}}$, which are distributions over hidden states and private action-observation histories, induced by the uninformed occupancy state $x$ and public observation history $h_{\mathrm{pub}} \in \mathcal{H}_{\mathrm{pub}}$; that is, for any hidden state $s$ and private histories $(h_1, h_2)$ of the two players: $o_{x,h_{\mathrm{pub}}}(s, h_1, h_2) \doteq \Pr(s, h_1, h_2 | \theta, h_{\mathrm{pub}})$. Uninformed occupancy states are convex combinations of these informed states, indexed by public observation histories. Letting $\boldsymbol{e}_{h_{\mathrm{pub}}}$ denote the one-hot vector for $h_{\mathrm{pub}}$, we have: $x = \sum_{h_{\mathrm{pub}} \in \mathcal{H}_{\mathrm{pub}}} \Pr(h_{\mathrm{pub}} | x) \cdot (o_{(x,h_{\mathrm{pub}})} \otimes \boldsymbol{e}_{h_{\mathrm{pub}}})$, where $o_{(x,h_{\mathrm{pub}})} \otimes \boldsymbol{e}_{h_{\mathrm{pub}}}$ denotes a Kronecker product. This decomposition allows the optimal value function $v_*: \mathcal{X} \to \mathbb{R}$ to be computed separately for each informed occupancy state, by selecting decision rules $d_{i,h_{\mathrm{pub}}} \in \mathcal{D}_{i,h_{\mathrm{pub}}}$ for player $i$ independently across all public observation histories $h_{\mathrm{pub}}$.

**Theorem 4.** *The optimal state-value function* $v_*: \mathcal{X} \to \mathbb{R}$ *of transition-independent zs-SG* $\mathcal{M}'$*, as defined by [Bellman](#)'s optimality equations in Theorem [3](#), is a linear map over informed occupancy states. Specifically, if* $x \in \mathcal{F}$*, then* $v_*(x) = 0$*; otherwise,*

$$v_*(x) = \sum_{h_{pub} \in \mathcal{H}_{pub}} \Pr(h_{pub} | x) \max_{d_{1,h_{pub}} \in \mathcal{D}_{1,h_{pub}}} \min_{d_{2,h_{pub}} \in \mathcal{D}_{2,h_{pub}}} q_*(o_{(x,h_{pub})}, d_{1,h_{pub}}, d_{2,h_{pub}})$$

$$q_*(o_{(x,h_{pub})}, d_{1,h_{pub}}, d_{2,h_{pub}}) = \rho(o_{(x,h_{pub})}, d_{1,h_{pub}}, d_{2,h_{pub}}) + \gamma v_*(\tau(o_{(x,h_{pub})}, d_{1,h_{pub}}, d_{2,h_{pub}})),$$

*where* $o_{(x,h_{pub})}$ *denotes the informed occupancy state induced by* $(x, h_{pub})$.

While the informed planner leverages structure across public observations, it remains agnostic to the private action-observation histories of each player. The *marginal planner* further refines this reasoning by branching on one player's private history. Specifically, for player $i$, the marginal planner operates over *marginal occupancy states* $c_{i,(x,h_{\mathrm{pub}},h_i)}$, which represent distributions over hidden states and the opponent private histories, conditioned on the uninformed occupancy states $x$, the public observation history $h_{\mathrm{pub}}$, and the private history $h_i$. That is, for hidden state $s$ and opponent private histories $h_{-i}$, one has $c_{i,(x,h_{\mathrm{pub}},h_i)}(s, h_{-i}) = \Pr(s, h_{-i} | x, h_{\mathrm{pub}}, h_i)$. Uninformed occupancy states can be expressed as convex combinations of these marginal states, indexed by public and private histories. Let $\boldsymbol{e}_{h_{\mathrm{pub}}}$ and $\boldsymbol{e}_{h_i}$ denote the one-hot vectors for $h_{\mathrm{pub}}$ and $h_i$, respectively. Then

$x = \sum_{h_{\text{pub}}} \Pr(h_{\text{pub}}|x) \sum_{h_i} \Pr(h_i|x, h_{\text{pub}}) \cdot \left( c_{i,(x,h_{\text{pub}},h_i)} \otimes e_{h_{\text{pub}}} \otimes e_{h_i} \right)$, where $\otimes$ denotes the Kronecker product. This refinement allows the marginal planner to isolate the strategic impact of private information while maintaining a full representation of the game evolution. As such, marginal planners are the most informed entities in the hierarchy, incorporating both public and private signals in their reasoning. This decomposition unveils that the optimal value function $v_* : \mathcal{X} \to \mathbb{R}$ is uniformly continuous across uninformed occupancy states.

**Theorem 5.** *The optimal state-value function $v_* : \mathcal{X} \to \mathbb{R}$ is uniformly continuous across uninformed occupancy states. There exists a collection $\Gamma_1$ of finite sets $\Gamma_2$ of functions $\alpha_2$, each linear over marginal occupancy states $c_2$, such that for any uninformed occupancy state $x$, we have:*

$$v_*(x) = \sum_{h_{pub} \in \mathcal{H}_{pub}} \Pr(h_{pub}|x) \left[ \max_{\Gamma_2 \in \Gamma_1} \sum_{h_2 \in \mathcal{H}_2} \Pr(h_2|h_{pub}, x) \min_{\alpha_2 \in \Gamma_2} \alpha_2(c_{2,(x,h_{pub},h_2)}) \right].$$

In practice, point-based methods approximate the optimal value function using a finite collection $\Gamma_1$ of finite sets $\Gamma_2$ of linear functions over sampled marginal occupancy states. This suffices to support value updates and policy extraction with performance guarantees, as formalised in Section 5.

# 5 $\varepsilon$-Optimally Solving $\mathcal{M}$ as $\mathcal{M}'$ via Point-Based Value Iteration

The hierarchy of planners offers more than structural insight—it enables practical computation. In particular, the uniform continuity of the value function, established at the level of the marginal planner, allows point-based representations to be leveraged without compromising $\varepsilon$-optimality. We exploit this structure to solve the reduced game $\mathcal{M}'$ using a point-based value iteration (PBVI) algorithm [Pineau et al., 2003], yielding an $\varepsilon$-optimal solution to the original zs-POSG $\mathcal{M}$. At the core of this method is the ability to define action-value functions $q_* : \mathcal{X} \times \mathcal{D}_1 \to \mathbb{R}$ over uninformed occupancy states, from which greedy decision rules are extracted via linear programming. These rules are then used to propagate and refine value estimates. The resulting value function helps extracting a robust focal policy whose exploitability is explicitly bounded in terms of the selected points.

To enable point-based backups, our PBVI Algorithm 1 variant samples a finite set of informed occupancy states $\mathcal{O}'$ that jointly induce a representative set of marginal occupancy states. The process begins with the initial informed state and expands the sample set by simulating one-step forward transitions. For each marginal state $c_2 \in \mathcal{C}'_2$, and for each focal decision rule $d_1$ and opponent action $a_2$, we compute a successor marginal state $\tau(c_2, d_1, a_2)$ and reconstruct compatible informed states by exploiting the convex decomposition linking marginal and informed occupancy states via public observation histories. The newly obtained marginal state is retained only if it lies farther—in $\ell_1$-norm—from the current sample set than any existing point, ensuring the sample density improves in worst-case regions. At each expansion, the marginal set grows by at most a factor of two. This synchronized sampling yields a nested hierarchy of representative informed and marginal occupancy states suitable for accurate, generalisable point-based value backups.

Given an uninformed occupancy state $x$ at stage $t$ and joint decision rules $(d_1, d_2)$, the expected cumulative discounted payoff under joint policies $(\pi_{1,\ell-t}, \pi_{2,\ell-t})$ is defined as $q_{\pi_{1,\ell-t},\pi_{2,\ell-t}}(x, d_1, d_2) = \rho(x, d_1, d_2) + \gamma v_{\pi_{1,\ell-t},\pi_{2,\ell-t}}(\tau(x, d_1, d_2))$. The optimal action-value function $q_*$ is given by $q_*(x, d_1) = \min_{d_2 \in \mathcal{D}_2} [\rho(x, d_1, d_2) + \gamma v_*(\tau(x, d_1, d_2))]$. The uniform continuity of $v_*$ ensures that $q_*$ generalises across nearby uninformed occupancy states.

**Corollary 1.** *The optimal action-value function $q_* : \mathcal{X} \times \mathcal{D}_1 \to \mathbb{R}$ is uniformly continuous across uninformed occupancy states. There exists a collection $\Phi_1$ of finite sets $\Phi_2$ of functions $\phi_2$, each linear over marginal occupancy states $c_2$ and private decision rules $d_1$. Thus, for any uninformed occupancy state $x$ and private decision rule $d_1$,*

$$q_*(x, d_1) = \sum_{h_{pub}} \Pr(h_{pub}|x) \max_{\phi_1 \in \Delta(\Phi_1)} \sum_{\Phi_2 \in \Phi_1} \phi_1(\Phi_2) \sum_{h_2} \Pr(h_2|x, h_{pub}) \min_{a_2, \phi_2 \in \Phi_2} \phi_2(c_{2,(x,h_{pub},h_2)}, d_1, a_2),$$

Point-based methods approximate the optimal action-value function using a finite collection $\Phi_1$ of finite sets $\Phi_2$ of linear functions $\phi_2$ over sampled marginal occupancy states and decision rules. We now describe how to extract a greedy decision rule for focal player 1 from the uniform continuity of action-value function $q$. Thanks to the uniform continuity of this function in informed occupancy states, this optimisation can be cast as a tractable linear program.

**Theorem 6.** *Let $o$ be an informed occupancy state. Then the decision rule $d_1$ maximising $q(o, \cdot)$ can be computed as the solution of the following linear program with:*

- *$O(|\Phi_1| \cdot |\mathcal{H}_1(o)| \cdot |\mathcal{A}_1|)$ **variables**,*
- *$O(|\Phi_1| \cdot |\Phi_2^*| \cdot |\mathcal{H}_2(o)| \cdot |\mathcal{A}_2|)$ **constraints**,*

*where $\Phi_2^*$ denotes the largest set of linear functions within any $\Phi_2 \in \Phi_1$. The linear program is:*

*Maximise* $\quad \sum_{\Phi_2 \in \Phi_1} \sum_{h_2 \in \mathcal{H}_2(o)} \Pr(h_2|o) \cdot \upsilon(h_2, \Phi_2)$

*Subject to* $\quad \sum_{a_1 \in \mathcal{A}_1} \sum_{\Phi_2 \in \Phi_1} \xi_1(a_1, \Phi_2|h_1) = 1, \quad \forall h_1 \in \mathcal{H}_1(o),$

$\qquad\qquad \upsilon(h_2, \Phi_2) \leq \sum_{h_1} \sum_{a_1} \xi_1(a_1, \Phi_2|h_1) \sum_{s \in \mathcal{S}} \phi_2(s, h_1, a_1, a_2) \cdot c_{2,(o,h_2)}(s, h_1),$

$\qquad\qquad \forall \Phi_2, \forall \phi_2 \in \Phi_2, \forall a_2 \in \mathcal{A}_2, \forall h_2 \in \mathcal{H}_2(o),$

*where $\mathcal{H}_i(o)$ denotes the finite set of private histories of player $i$ reachable in $o$. The variable $\xi_1(a_1, \Phi_2|h_1)$ encodes the probability of taking action $a_1$ in history $h_1$, assuming the value model $\Phi_2$ is drawn from $\phi_1$. The inner constraint ensures that the worst-case evaluation $\upsilon(h_2, \Phi_2)$ is always pessimistic—i.e., no matter how the opponent reacts, the value function bound holds.*

This primal linear program computes solution $\xi_1$, which induces a decision rule $d_1$ that is robust to all potential responses from player 2. Solving the primal linear program identifies the safest and most effective decision rule $d_1$ under this structure. The solution obtained from the primal linear program can be used to improve the current estimate of the value function. The following result formalises this improvement: the update operator raises the value at least at one informed occupancy state while preserving or improving it elsewhere. This monotonicity ensures progress with each update, forming the foundation of convergence guarantees in point-based dynamic programming.

**Corollary 2.** *Let $v$ and $q$ be the current state- and action-value functions represented by finite collections $\Gamma_1$ of sets $\Gamma_2$, and $\Phi_1$ of sets $\Phi_2$, respectively. Let $o$ be an informed occupancy state, and let $\xi_1$ denote the solution of the greedy linear program from Theorem 6 at $o$. We define an updated value function $v\prime$ by augmenting $\Gamma_1$ with a new set $\Gamma_{2,(c_2', \xi_1)}$ of linear functions $\alpha_{2,(c_2)}$ given by:*

$$\alpha_{2,(c_2)} = \sum_{\Phi_2 \in \Phi_1} \operatorname{argmin}_{\alpha_2^{\phi_2, a_2} \,:\, \phi_2 \in \Phi_2, \, a_2 \in \mathcal{A}_2} \alpha_2^{\phi_2, a_2}(c_2)$$

$$\alpha_2^{\phi_2, a_2}(s, h_1) = \sum_{a_1} \xi_1(a_1, \Phi_2|h_1) \cdot \phi_2(s, h_1, a_1, a_2).$$

*Then $v\prime(x) \geq v(x)$ for any uninformed occupancy state $x$ induced by $\mathcal{C}_2'$, and $v\prime(x) > v(x)$ for at least one such $x$ if the greedy update yields a strict improvement.*

To further improve scalability, we incorporate two distinct pruning strategies: one that removes dominated elements from the set of linear functions $\Gamma_2$ (*cf.* Algorithm 2) and another that discards redundant informed occupancy states from the sample set $\mathcal{O}'$ (*cf.* Algorithm 3) . Each pruning operation introduces approximation error in the value function. While these errors can be controlled individually, combining both strategies may lead to compounding errors and the loss of formal performance guarantees.

We now present a bound on the exploitability of the focal policy computed by our point-based value iteration algorithm in the finite-horizon setting of length $\ell$. Given any sample set $\mathcal{C}_{2,0:\ell}'$, the algorithm produces a focal policy $\pi_1'$ with estimated value $v_1(b)$. The exploitability of this policy is defined as $\varepsilon \doteq v_{1,*}(b) - \min_{\pi_2' \in \Pi_2} v_{\pi_1', \pi_2'}(b)$, and quantifies the worst-case suboptimality of $\pi_1'$ against a best-responding opponent. The exploitability decreases as the sampled set $\mathcal{C}\prime_{2,0:\ell}$ becomes denser in the marginal occupancy space; in the limit, $v_1(b)$ converges to the optimal value $v_{1,*}(b)$, and $\varepsilon$ approaches zero. The remainder of this section formalises and proves this bound. To this end, we define the density $\delta$ as the maximum distance from any reachable marginal occupancy state to the sample set $\mathcal{C}\prime_{2,0:\ell}$; more precisely, $\delta \doteq \max_{t=0,\dots,\ell} \max_{c_2 \in \mathcal{C}_{2,t}} \min_{c\prime_2 \in \mathcal{C}\prime_{2,t}} \|c_2 - c_2'\|_1$ . Let $m > 0$ be a constant such that $\|r\|_\infty \leq m$.

**Theorem 7.** *For any marginal occupancy sample sets $\mathcal{C}\prime_{2,0:\ell}$, the exploitability of the focal policy obtained via PBVI and evaluated at the initial state distribution, is bounded as*

$$\varepsilon \leq \frac{4m\delta}{(1-\gamma)^2} \cdot [1 + (\ell+1)\gamma^{\ell+2} - (\ell+2)\gamma^{\ell+1}].$$

It is worth noticing that whenever $\ell$ goes to infinity, our exploitability bound is twice the error-bound from Pineau et al. [2003] for infinite-horizon partially observable Markov decision processes.

# 6  Empirical Evaluation

We evaluate our method on a suite of established benchmarks for simultaneous-move partially observable stochastic games (POSGs): Adversarial Tiger, Competitive Tiger, Recycling, Mabc, Matching Pennies, and three Pursuit-Evasion variants. These benchmarks are among the most challenging in the POSG literature; see http://masplan.org/ for detailed descriptions. Several were originally common-payoff problems and have been adapted to the competitive setting by reversing the objective for player 2. For each benchmark, we compare three variants of our PBVI algorithm: $PBVI_1$ (baseline, without pruning), $PBVI_2$, and $PBVI_3$ (both applying the bounded pruning scheme from Section 5). We benchmark against the HSVI implementation of Delage et al. [2023] and the CFR+ algorithm of Tammelin [2014]. Table 3 summarises results for the most computationally demanding horizons, reporting the final value reached by each algorithm and the exploitability of the resulting focal policy. Full results for all tested horizons $\ell \in \{2, 3, 4, 5, 7, 10\}$ are deferred to Table 3. To foster reproducibility, the full codebase, including configuration files and experimental scripts, is available at Escudie et al. [2025].

**Exploitability & $\varepsilon$-Optimal Values:**   Table 3 presents the algorithms achieving the lowest exploitability in **magenta**. Across all benchmarks, there always exists at least one PBVI variant that significantly outperforms both HSVI and CFR+. Among these, $PBVI_3$ emerges as the most reliable, consistently yielding the lowest exploitability except for horizons $\ell \in \{4, 5\}$ on the Competitive Tiger benchmark and $\ell = 7$ on Mabc. Nonetheless, $PBVI_1$ and $PBVI_2$ also perform favourably, outperforming both baselines in nearly every instance. Notably, CFR+ fails to scale in most cases, running out of memory on many benchmarks, while HSVI frequently exceeds the time limit. This behaviour reflects fundamental limitations of both methods: HSVI's backup operator grows exponentially with the horizon $\ell$, unless the problem exhibits strong structure—explaining its poor scalability beyond small-horizon settings, with Matching Pennies being a notable exception; CFR+, in contrast, is sensitive to the size of the history space, which explains why it performs well on compact extensive-form games such as Matching Pennies at $\ell = 10$, but fails on shallow instances like Competitive Tiger at $\ell = 4$, where the set of local histories is already large. Regarding the values achieved, we observe only minor differences between PBVI variants, with most converging to nearly identical solutions. Discrepancies occur primarily on the Competitive Tiger and Pursuit-Evasion benchmarks, where the variants exhibit slightly divergent convergence behaviours.

Table 1: Snapshot of empirical results. Games are ordered by increasing planning horizon $\ell$, and within each horizon by ascending number of local histories. For each setting, we report the value $v(b)$ and exploitability $\varepsilon$. OOT indicates a timeout (2-hour limit), OOM denotes out-of-memory runs, and '–' means the exploitability budget was exceeded. Best results are highlighted in **magenta**.

| Game ($\ell$) | $PBVI_1$ $v(b)$ | $\varepsilon$ | $PBVI_2$ $v(b)$ | $\varepsilon$ | $PBVI_3$ $v(b)$ | $\varepsilon$ | HSVI [Delage et al., 2023] $v(b)$ | $\varepsilon$ | CFR+ [Tammelin, 2014] $v(b)$ | $\varepsilon$ |
|---|---|---|---|---|---|---|---|---|---|---|
| pursuit-evasion-2x2(2) | **0.00** | **0.00** | **0.00** | **0.00** | **0.00** | **0.00** | **0.00** | **0.00** | **0.00** | 0.08 |
| pursuit-evasion-3x3x2(2) | **0.00** | **0.00** | **0.00** | **0.00** | **0.00** | **0.00** | **0.00** | **0.00** | **0.00** | 0.01 |
| pursuit-evasion-3x3x1(2) | **0.00** | **0.00** | **0.00** | **0.00** | **0.00** | **0.00** | **0.00** | **0.00** | **0.00** | 0.01 |
| pursuit-evasion-2x2(3) | **0.00** | **0.00** | **0.00** | **0.00** | **0.00** | **0.00** | **0.00** | **0.00** | **0.00** | 0.06 |
| pursuit-evasion-3x3x2(3) | -22.00 | 0.92 | -41.00 | 3.86 | **0.00** | **0.00** | OOT | | **0.00** | 0.10 |
| pursuit-evasion-3x3x1(3) | **0.00** | **0.00** | **0.00** | **0.00** | **0.00** | **0.00** | OOT | | **0.00** | 0.17 |
| matching-pennies(4) | **0.60** | 0.04 | **0.60** | 0.08 | **0.60** | 0.01 | 0.60 | 0.01 | **0.60** | 0.01 |
| adversarial-tiger(4) | **-0.76** | **0.00** | **-0.76** | **0.00** | **-0.76** | **0.00** | OOT | | -0.75 | 0.01 |
| mabc(4) | **0.11** | **0.00** | **0.11** | 0.00 | **0.11** | **0.00** | OOT | | **0.11** | 0.00 |
| recycling(4) | **0.36** | 0.01 | **0.36** | 0.01 | **0.36** | **0.00** | OOT | | **0.36** | 0.03 |
| competitive-tiger(4) | **-0.03** | 0.03 | -0.07 | **0.00** | -0.05 | 0.03 | OOT | | OOM | |
| pursuit-evasion-2x2(4) | -22.00 | 37.00 | -26.00 | 48.00 | -28.00 | **1.00** | OOT | | OOM | |
| pursuit-evasion-3x3x2(4) | **0.00** | 95.00 | -6.00 | 9.00 | **0.00** | 6.00 | OOT | | OOM | |
| pursuit-evasion-3x3x1(4) | **0.00** | 6.00 | -6.00 | 6.00 | **0.00** | **0.00** | OOT | | OOM | |
| matching-pennies(5) | **0.80** | 0.01 | 0.78 | 0.01 | 0.78 | **0.00** | **0.80** | 0.01 | **0.80** | 0.01 |
| adversarial-tiger(5) | **-0.95** | 0.04 | **-0.95** | 0.01 | **-0.95** | **0.00** | OOT | | -0.95 | 0.03 |
| mabc(5) | **0.12** | **0.00** | **0.12** | 0.01 | **0.12** | **0.00** | OOT | | **0.12** | 0.01 |
| recycling(5) | **0.40** | 0.01 | **0.40** | 0.05 | **0.40** | 0.01 | OOT | | OOM | |
| competitive-tiger(5) | **-0.06** | 0.01 | -0.08 | **0.00** | -0.10 | 0.02 | OOT | | OOM | |
| matching-pennies(7) | **1.20** | 0.05 | **1.20** | 0.04 | 1.19 | **0.04** | OOT | | **1.20** | 0.06 |
| adversarial-tiger(7) | **-1.40** | **0.00** | **-1.40** | 0.09 | **-1.40** | **0.00** | OOT | | OOM | |
| mabc(7) | **0.14** | **0.00** | **0.14** | 0.00 | **0.14** | 0.00 | OOT | | OOM | |
| recycling(7) | **0.51** | 0.07 | 0.50 | 0.04 | 0.49 | **0.02** | OOT | | OOM | |
| competitive-tiger(7) | **-0.15** | 0.03 | -0.17 | 0.04 | **-0.15** | **0.02** | OOT | | OOM | |
| matching-pennies(10) | **1.80** | – | **1.80** | – | **1.80** | – | OOT | | **1.80** | 0.06 |
| adversarial-tiger(10) | **-2.00** | – | **-2.00** | – | **-2.00** | – | OOT | | OOM | |
| mabc(10) | **0.17** | – | 0.18 | – | 0.20 | – | OOT | | OOM | |
| recycling(10) | **0.60** | – | **0.60** | – | **0.60** | – | OOT | | OOM | |
| competitive-tiger(10) | OOT | | -0.29 | – | **-0.20** | – | OOT | | OOM | |

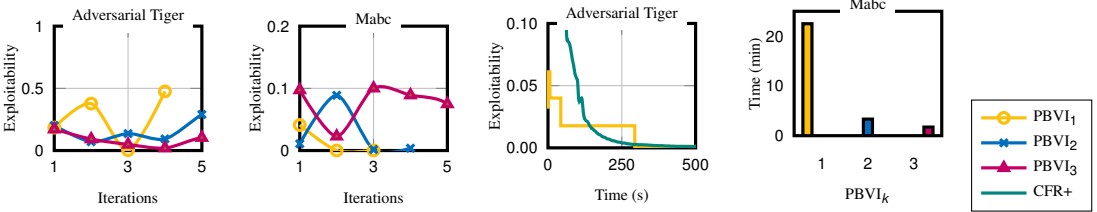

Figure 3: Exploitability of $PBVI_k$ across iterations and runtime on Adversarial Tiger and Mabc ($\ell = 5$), with CFR+ for comparison. Rightmost plot shows time-to-convergence for $PBVI_k$ on Mabc.

**Additional Results & Insights:** We now discuss complementary insights that characterise the empirical behaviour of our PBVI variants, see Apprendix E.2. In terms of scalability with respect to the planning horizon $\ell$, many PBVI versions remain tractable across all tested instances, including large horizons up to $\ell = 10$, without running OOT or OOM. This is enabled by the pruning heuristics introduced in Section 5, which become essential as the naive $PBVI_1$ variant grows increasingly expensive. The bounded pruning mechanisms in $PBVI_2$ and $PBVI_3$ allow deeper backups and longer runs, as illustrated in the first two panels of Figure 3 for the Adversarial Tiger and Mabc benchmarks with $\ell = 5$. In addition, our PBVI variants converge significantly faster than HSVI and CFR+ across most benchmarks. The third panel of Figure 3 shows $PBVI_1$ achieving a low-exploitability solution substantially earlier than CFR+ on Adversarial Tiger with $\ell = 5$. For further convergence statistics and runtime comparisons, we refer the reader to Table 3 in the supplemental material. We conclude by highlighting the trade-off introduced by pruning. Although it weakens theoretical guarantees, the empirical gains in efficiency are substantial. As shown in the final panel of Figure 3, $PBVI_2$ and $PBVI_3$ solve Mabc with $\ell = 5$ in a fraction of the time required by $PBVI_1$, while maintaining comparably low exploitability as confirmed in Table 3.

## 7 Conclusion

We introduced the first principled and lossless reduction from zs-POSGs to transition-independent zs-SGs. While transition independence has been applied in common-payoff POSGs [Becker et al., 2003, 2004, Dibangoye et al., 2012, 2013b, 2014], this work is the first to extend it to adversarial games, providing both theoretical guarantees and scalable planning methods. By exploiting a hierachy of planners and the uniform continuity of the value function, we developed a point-based value iteration algorithm that operates over a structured sample of marginal and informed occupancy states. The method supports value-function improvement via linear programming, admits an explicit exploitability bound, and scales to challenging benchmarks previously beyond reach for dynamic programming theory and algorithms. Our results demonstrate both the theoretical soundness and practical viability of planning with occupancy-set dependent policies in adversarial settings with imperfect information. Together, these contributions establish a general pathway for transferring dynamic programming theory and algorithms from stochastic games to partially observable settings, and offer a promising foundation for unifying the solution of cooperative, competitive, and mixed-motive POSGs under a common framework. Such unification is groundbreaking because it dissolves the long-standing divide between algorithmic principles across multi-agent problems, enabling a shared planning infrastructure that can adapt flexibly to diverse strategic interactions and uncertainty structures.

**Limitations.** While the linear programs generated by our method are significantly smaller than those used in HSVI-based approaches, their size still grows with the planning horizon in the worst case, potentially limiting scalability. Future work could explore compact representations or local update schemes, such as regret minimisation, to mitigate this bottleneck.

## Acknowledgments

This work was supported by the French National Research Agency (ANR) under projects ANR-19-CE23-0018 (Planning and Learning to Act in Systems of Multiple Agents), ANR-19-CE23-0006 (Data and Prior: Machine Learning and Control), and ANR-21-CE23-0016 (Multi-Agent Trust

Decision Process for the Internet of Things). The authors also acknowledge financial support for Erwan C. Escudie through a PhD scholarship from the University of Groningen.

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

# A Preliminaries

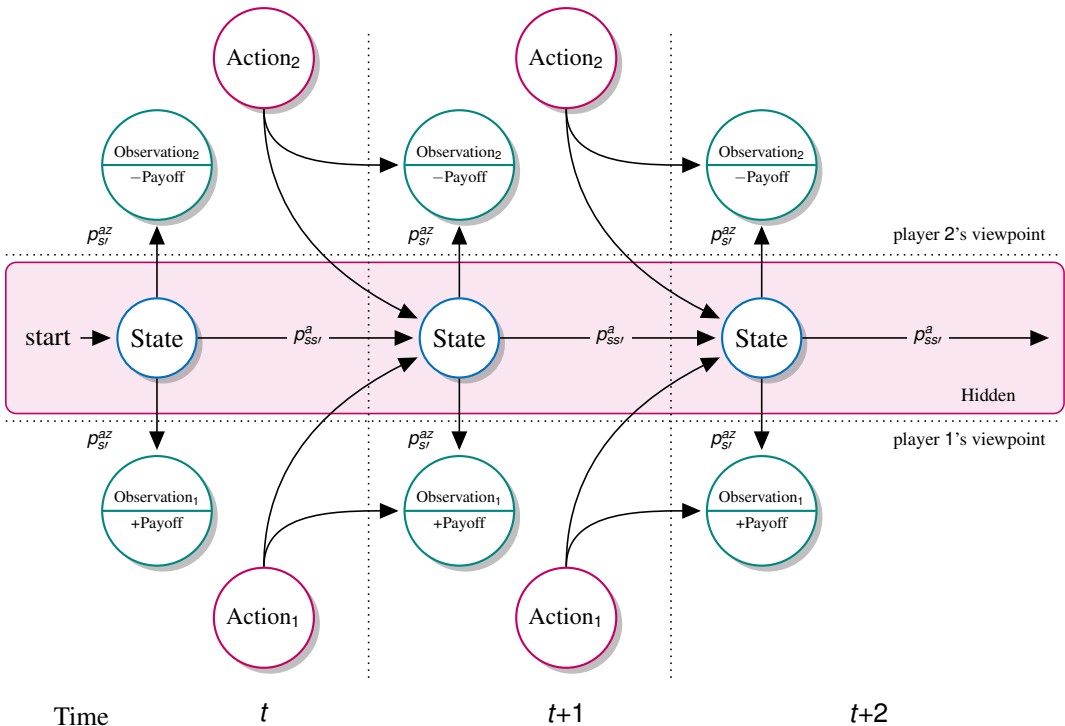

Figure 4: A graphical model of a two-player zero-sum partially observable stochastic game. Each triple $z \doteq (z_1, z_2, w)$ comprises private and public observations. The diagram illustrates an influence process over three stages: central nodes represent the hidden states ($s_t$); the top and bottom rows show the private observations and actions of players 2 and 1, respectively. Observation nodes also include the local payoff: "+" denotes a gain for player 1, and "−" a loss for player 2. Directed edges indicate probabilistic dependencies: actions influence transitions and observations, while observations inform future actions. The shaded region highlights the hidden environment state from each player's viewpoint, emphasising the decentralised and asymmetric information structure. This diagram captures the sequential, partially observable, and adversarial nature of zs-POSGs. The underlying dynamics decompose into two functions, the state transition matrices $\{p_{ss'}^a\}$ and the observation matrices $\{p_{s'}^{az}\}$, where $p(s', z|s, a) = p_{ss'}^a \cdot p_{s'}^{az}$.

# B Reducing zs-POSGs to Transition-Independent zs-SGs

## B.1 Proof of Lemma 1

**Lemma 1.** *The reduced game $\mathcal{M}'$ admits a well-defined value $v_*'(b)$, which satisfies the minimax identity:* $v_*'(b) = \min_{\psi_2 \in \Psi_2} \max_{\psi_1 \in \Psi_1} v_{\psi_1, \psi_2}'(b) = \max_{\psi_1 \in \Psi_1} \min_{\psi_2 \in \Psi_2} v_{\psi_1, \psi_2}'(b).$

*Proof.* The proof shows that any occupancy-set dependent policy induces a valid behavioural strategy, thereby enabling the application of the minimax theorem. Let $\psi_i \colon x_{i,t} \mapsto d_{i,t}$ be an occupancy-set dependent policy for player $i$, assigning a decision rule at each occupancy set $x_{i,t}$. This induces a behavioural strategy $\pi_i$ in the extensive-form game associated with the zs-POSG. The construction is recursive. Let $x_{i,0}$ denote the initial occupancy set induced by the prior $\{b\}$. At each stage $t = 0, \ldots, \ell$, define: $d_{i,t} \doteq \psi_i(x_{i,t})$, and for all $(h_{i,t}, h_{\mathrm{pub},t}) \in \mathcal{H}_{i,t} \times \mathcal{H}_{\mathrm{pub},t}$, define: $\pi_i(\cdot|h_{i,t}, h_{\mathrm{pub},t}) \doteq d_{i,t}(\cdot|h_{i,t}, h_{\mathrm{pub},t})$. The next occupancy set is obtained by the deterministic transition $x_{i,t+1} = \tau_i(x_{i,t}, d_{i,t})$. This construction yields a behavioural strategy $\pi_i$ that is fully defined across all stages, consistent with the player's local information, and realisable in the extensive-form game associated with the zs-POSG. Since the game has perfect recall, the equivalence between behavioural and mixed strategies

holds [Kuhn, 1953], and thus the minimax theorem [Neumann, 1928, Sion, 1958] applies to the reduced game $\mathcal{M}'$ by turning it into a normal-form game where actions are pure strategies–also known as deterministic history-dependent policies [Delage et al., 2023]. This concludes the proof. $\qquad\square$

### B.2 Proof of Theorem 1

**Theorem 1.** *The reduced game $\mathcal{M}'$ constitutes a lossless reduction of the original zs-POSG $\mathcal{M}$.*

*Proof.* The proof verifies that the reduced game $\mathcal{M}'$ satisfies the three criteria of a lossless reduction from the original zs-POSG $\mathcal{M}$. Let $(\psi_1, \psi_2)$ be a joint policy in the reduced game. The value of $\mathcal{M}'$ under this policy, starting from the initial uninformed occupancy state $x_0 = b$, is given by:

$$v'_{\psi_1,\psi_2}(b) \doteq \sum_{t=0}^{\ell} \gamma^t \cdot \rho(x_t, d_{1,t}, d_{2,t}) | x_t = \varphi(x_{1,t}, x_{2,t}), d_{i,t} = \psi_i(x_{i,t}), x_{i,t} = \tau_i(x_{i,t-1}, d_{i,t-1})$$

$$= \sum_{t=0}^{\ell} \gamma^t \cdot \mathbb{E}_{(s_t, a_{1,t}, a_{2,t}) \sim \Pr(\cdot | x_t, d_{1,t}, d_{2,t})}[r(s_t, a_{1,t}, a_{2,t})] \tag{1}$$

$$= \sum_{t=0}^{\ell} \mathbb{E}_{(s_0,\ldots,s_\ell, a_{1,0:\ell}, a_{2,0:\ell}) \sim \Pr(\cdot | b, d_{1,0:\ell}, d_{2,0:\ell})}[\gamma^t \cdot r(s_t, a_{1,t}, a_{2,t})] \tag{2}$$

$$= \sum_{t=0}^{\ell} \mathbb{E}_{(s_0,\ldots,s_\ell, a_{1,0:\ell}, a_{2,0:\ell}) \sim \Pr(\cdot | b, \pi_1, \pi_2)}[\gamma^t \cdot r(s_t, a_{1,t}, a_{2,t})] \doteq v_{\pi_1,\pi_2}(b) \tag{3}$$

Equations (1)–(3) follow from the definition of $\rho$, linearity of expectation, and the mapping $\pi_i = (d_{i,0}, \ldots, d_{i,\ell})$ induced by $\psi_i$. This establishes value preservation.

If we let $(\psi_1^*, \psi_2^*)$ be an optimal strategy in $\mathcal{M}'$, then for any $\psi_i \in \Psi_i$,

$$v'_{\psi_1,\psi_2^*}(b) \leq v'_{\psi_1^*,\psi_2^*}(b) \leq v'_{\psi_1^*,\psi_2}(b). \tag{4}$$

Let $\psi_i(x_{i,t}) = d_{i,t}$, with $d_{i,t}(\cdot | h_{i,t}) = \pi_i(\cdot | h_{i,t})$, where $x_{i,t}$ is the occupancy set summarising $(d_{i,0}, \ldots, d_{i,t-1})$. Then, by (3) and (4), for any $\pi_i \in \Pi_i$,

$$v_{\pi_1,\pi_2^*}(b) \leq v_{\pi_1^*,\pi_2^*}(b) \leq v_{\pi_1^*,\pi_2}(b), \tag{5}$$

which confirms equilibrium correspondence.

Conversely, any joint policy $(\pi_1, \pi_2)$ in the original game induces decision-rule sequences $(d_{i,0}, \ldots, d_{i,\ell})$, which define an occupancy-set dependent policy $\psi_i$ such that $\psi_i(x_{i,t}) = d_{i,t}$. By the same construction, $v'_{\psi_1,\psi_2}(b) = v_{\pi_1,\pi_2}(b)$, which ensures that all equilibria and values in the original game are preserved in the reduced game.

Finally, the definition of admissible decision rules $\widetilde{\mathcal{D}}_{i,t}$ for $\mathcal{M}'$ at each stage $t$ and player $i$ remains unchanged with respect to admissible decision rules for the original game $\mathcal{M}$:

$$\widetilde{\mathcal{D}}_{i,t} = \mathcal{D}_{i,t}. \tag{6}$$

The reduction preserves the original observation structure and introduces no informational asymmetry, ensuring that the information structure is preserved.

Combining (3), (5), and (6), we conclude that $\mathcal{M}'$ is a lossless reduction of $\mathcal{M}$. $\qquad\square$

## C Solving Transition-Independent zs-SGs via A Hierarchy of Planners

### C.1 Proof of Theorem 2

**Theorem 2.** *The optimal state-value function $v_{i,*}: \mathcal{X}_i \to \mathbb{R}$ of $\mathcal{M}_i$ satisfies Bellman's optimality equations: $v_{i,*}(x_i) = \rho_i(x_i)$ if $x_i \in \mathcal{F}_i$, and $v_{i,*}(x_i) = \mathrm{opt}_{d_i \in \mathcal{D}_i} v_{i,*}(\tau_i(x_i, d_i))$ otherwise; with an optimal policy given by $\psi_i^*: x_i \mapsto \arg\mathrm{opt}_{d_i \in \mathcal{D}_i} v_{i,*}(\tau_i(x_i, d_i))$, where the optimisation operator $\mathrm{opt}$ corresponds to $\max$ for player 1 and $\min$ for player 2.*

*Proof.* We prove the theorem by induction on the number of remaining stages until the planning horizon $\ell + 1$, starting from a focal occupancy set $x_{i,t} \in \mathcal{X}_i$.

**Base case (stage $t = \ell + 1$):** At the final decision stage, the focal planner reaches a terminal occupancy set $x_{i,\ell+1} \in \mathcal{F}_i$. By definition, the expected return from this state is the terminal reward:

$$v_{i,*}(x_{i,\ell+1}) = \rho_i(x_{i,\ell+1}),$$

which matches the first part of the optimality equations.

**Inductive step:** Suppose Bellman's optimality equation holds at stage $t + 1$ for all $x_{i,t+1} \in \mathcal{X}_i$, i.e.,

$$v_{i,*}(x_{i,t+1}) = \begin{cases} \rho_i(x_{i,t+1}) & \text{if } x_{i,t+1} \in \mathcal{F}_i, \\ \mathrm{opt}_{d_{i,t+1} \in \mathcal{D}_i} v_{i,*}(\tau_i(x_{i,t+1}, d_{i,t+1})) & \text{otherwise.} \end{cases}$$

Now consider stage $t$. The planner must choose a decision rule $d_{i,t} \in \mathcal{D}_i$, transitioning to $x_{i,t+1} = \tau_i(x_{i,t}, d_{i,t})$, and then continuing optimally from there. The value of applying the optimal continuation policy in policy space $\Pi_{i,t}$ from occupancy set $x_{i,t}$ is:

$$\begin{aligned} v_{i,*}(x_{i,t}) &= \mathrm{opt}_{\psi_{i,t} \in \Psi_{i,t}} v_{i,\psi_{i,t}}(x_{i,t}) \\ &= \mathrm{opt}_{d_{i,t} \in \mathcal{D}_{i,t}} \mathrm{opt}_{\psi_{i,t+1} \in \Psi_{i,t+1}} v_{i,\psi_{i,t+1}}(\tau_i(x_{i,t}, d_{i,t})) \\ &= \mathrm{opt}_{d_{i,t} \in \mathcal{D}_{i,t}} v_{i,*}(\tau_i(x_{i,t}, d_{i,t})), \end{aligned}$$

which proves the recursive part of Bellman's optimality equation.

**Conclusion:** By induction, Bellman's optimality equations hold at all stages $t = \ell, \ell - 1, \dots, 0$. The greedy policy $\psi_i^*(x_{i,t}) \in \arg \mathrm{opt}_{d_{i,t} \in \mathcal{D}_{i,t}} v_{i,*}(\tau_i(x_{i,t}, d_{i,t}))$ selects the optimising decision rule at each stage and is therefore optimal. □

## C.2 Proof of Theorem 3

**Theorem 3.** *The optimal state-value function $v_* \colon \mathcal{X} \to \mathbb{R}$ of $\mathcal{M}'$ satisfies Bellman's optimality equations: $v_*(x) = 0$ if $x \in \mathcal{F}$, and $v_*(x) = \max_{d_1 \in \mathcal{D}_1} \min_{d_2 \in \mathcal{D}_2} [\rho(x, d_1, d_2) + \gamma v_*(\tau(x, d_1, d_2))]$ otherwise. For player $i$, the value of their focal planner at occupancy set $x_i$ at stage $t$ is given by:*

$$v_{i,*}(x_i) = \overline{\mathrm{opt}}_{x \in x_i} [g(x) + \gamma^t v_*(x)], \quad \forall x_i \in \mathcal{X}_i.$$

*Proof.* We proceed by induction on the number of remaining stages until the horizon $\ell + 1$, starting from any uninformed occupancy state $x \in \mathcal{X}$.

**Base case ($t = \ell + 1$):** By construction, all terminal states $x \in \mathcal{F}$ yield no further payoff. Thus, the expected cumulative reward is zero: $v_*(x) = 0$, for all $x \in \mathcal{F}$.

**Inductive step:** Assume that the optimal value satisfies Bellman's equation at stage $t + 1$, i.e.,

$$v_*(x) = \max_{d_1 \in \mathcal{D}_1} \min_{d_2 \in \mathcal{D}_2} [\rho(x, d_1, d_2) + \gamma v_*(\tau(x, d_1, d_2))] \quad \text{for all } x \text{ with } \ell - t - 1 \text{ stages to go.}$$

Now consider an uninformed occupancy state $x$ at stage $t$. The planner selects a joint decision rule $(d_1, d_2)$, leading deterministically to the next state $\tau(x, d_1, d_2)$. The expected cumulative reward is the sum of the immediate stage return and the discounted future value:

$$\begin{aligned} v_*(x) &= \max_{\pi_{1,t} \in \Pi_{1,t}} \min_{\pi_{2,t} \in \Pi_{2,t}} v_{\pi_{1,t}, \pi_{2,t}}(x), \quad \text{(by definition)} \\ &= \max_{d_{1,t} \in \mathcal{D}_{1,t}} \max_{\pi_{1,t+1} \in \Pi_{1,t+1}} \min_{d_{2,t} \in \mathcal{D}_{2,t}} \min_{\pi_{2,t+1} \in \Pi_{2,t+1}} v_{\pi_{1,t}, \pi_{2,t}}(x), \quad \text{(split } \pi_{i,t} = (d_{1,t}, \pi_{1,t+1})) \\ &\overset{\text{Delage et al. [2023, Thm. 3.2]}}{=} \max_{d_{1,t} \in \mathcal{D}_{1,t}} \min_{d_{2,t} \in \mathcal{D}_{2,t}} \min_{\pi_{2,t+1} \in \Pi_{2,t+1}} \max_{\pi_{1,t+1} \in \Pi_{1,t+1}} v_{\pi_{1,t}, \pi_{2,t}}(x), \quad \text{(swap min and max)} \\ &\overset{\text{Delage et al. [2023, Thm. 3.2]}}{=} \max_{d_{1,t} \in \mathcal{D}_{1,t}} \min_{d_{2,t} \in \mathcal{D}_{2,t}} \max_{\pi_{1,t+1} \in \Pi_{1,t+1}} \min_{\pi_{2,t+1} \in \Pi_{2,t+1}} v_{\pi_{1,t}, \pi_{2,t}}(x), \quad \text{(swap min and max)} \\ &= \max_{d_{1,t} \in \mathcal{D}_{1,t}} \min_{d_{2,t} \in \mathcal{D}_{2,t}} \max_{\pi_{1,t+1} \in \Pi_{1,t+1}} \min_{\pi_{2,t+1} \in \Pi_{2,t+1}} \left( \rho(x, d_1, d_2) + \gamma v_{\pi_{1,t+1}, \pi_{2,t+1}}(\tau(x, d_1, d_2)) \right) \\ &= \max_{d_{1,t} \in \mathcal{D}_{1,t}} \min_{d_{2,t} \in \mathcal{D}_{2,t}} \left( \rho(x, d_1, d_2) + \max_{\pi_{1,t+1} \in \Pi_{1,t+1}} \min_{\pi_{2,t+1} \in \Pi_{2,t+1}} \gamma v_{\pi_{1,t+1}, \pi_{2,t+1}}(\tau(x, d_1, d_2)) \right) \\ &= \max_{d_{1,t} \in \mathcal{D}_{1,t}} \min_{d_{2,t} \in \mathcal{D}_{2,t}} \rho(x, d_1, d_2) + \gamma v_*(\tau(x, d_1, d_2)), \end{aligned}$$

which confirms the recursive Bellman's optimality equations for the uninformed planner.

For each player $i$, recall that the value of $\mathcal{M}_i$ at occupancy set $x_i$ at stage $t$ is defined as the worst-case expected return under any possible realisation of the opponent's strategy:

$$v_{i,*}(x_i) = \mathrm{opt}_{\pi_{i,t}\in\Pi_{i,t}} v_{i,\pi_{i,t}}(x_i), \quad \text{(by definition)}$$
$$= \mathrm{opt}_{\pi_{i,t}\in\Pi_{i,t}} \rho_i(x_{i,\ell+1})|x_{i,\ell+1} = \tau_i(\tau_i(\dots \tau_i(\tau_i(x_i, d_{i,t}), d_{i,t+1})\dots), d_{i,\ell})$$
$$= \mathrm{opt}_{\pi_{i,t}\in\Pi_{i,t}} \overline{\mathrm{opt}}_{x_{\ell+1}\in x_{i,\ell+1}} g(x_{\ell+1})$$
$$= \mathrm{opt}_{\pi_{i,t}\in\Pi_{i,t}} \overline{\mathrm{opt}}_{x_{\ell+1}\in\tau_i(x_{i,\ell},d_{i,\ell})} g(x_{\ell+1})$$
$$= \mathrm{opt}_{\pi_{i,t}\in\Pi_{i,t}} \overline{\mathrm{opt}}_{x_{\ell}\in x_{i,\ell}} \overline{\mathrm{opt}}_{d_{-i,\ell}\in\mathcal{D}_{-i,\ell}} g(\tau(x_\ell, d_{i,\ell}, d_{-i,\ell}))$$
$$= \mathrm{opt}_{\pi_{i,t}\in\Pi_{i,t}} \overline{\mathrm{opt}}_{x_{\ell}\in x_{i,\ell}} \overline{\mathrm{opt}}_{d_{-i,\ell}\in\mathcal{D}_{-i,\ell}} [g(x_\ell) + \gamma^\ell \cdot \rho(x_\ell, d_{i,\ell}, d_{-i,\ell})]$$
$$= \mathrm{opt}_{\pi_{i,t}\in\Pi_{i,t}} \overline{\mathrm{opt}}_{x_{\ell}\in x_{i,\ell}} \overline{\mathrm{opt}}_{d_{-i,\ell}\in\mathcal{D}_{-i,\ell}} [g(x_\ell) + \gamma^\ell \cdot v'_{d_{i,\ell},d_{-i,\ell}}(x_\ell)]$$
$$= \mathrm{opt}_{\pi_{i,t}\in\Pi_{i,t}} \overline{\mathrm{opt}}_{x\in x_i} \overline{\mathrm{opt}}_{\pi_{-i,t}\in\Pi_{-i,t}} [g(x) + \gamma^t \cdot v'_{\pi_{i,t},\pi_{-i,t}}(x)]. \quad \text{(re-arranging terms)}$$

Observe that any two min or two max operators can be swapped freely. Moreover, a min and a max operator can be interchanged when both appear in the innermost (rightmost) positions. With appropriate ordering of operations, this flexibility allows arranging the optimisation operators in any desired sequence. It then follows that:

$$v_{i,*}(x_i) = \overline{\mathrm{opt}}_{x\in x_i} g(x) + \gamma^t \mathrm{opt}_{\pi_{i,t}\in\Pi_{i,t}} \overline{\mathrm{opt}}_{\pi_{-i,t}\in\Pi_{-i,t}} v'_{\pi_{i,t},\pi_{-i,t}}(x), \quad \text{(swap opt and } \overline{\mathrm{opt}})$$
$$= \overline{\mathrm{opt}}_{x\in x_i} [g(x) + \gamma^t v_*(x)],$$

where $\overline{\mathrm{opt}}$ is min for $i = 1$ and max for $i = 2$. This concludes the connection between the focal value function and that of the uninformed planner.

**Conclusion:** By induction, $v_*$ satisfies Bellman's optimality equations across $\mathcal{X}$, and induces the focal planners' values as minimax aggregations over their respective occupancy sets. $\square$

### C.3   Proof of Theorem 4

**Theorem 4.** *The optimal state-value function $v_* : \mathcal{X} \to \mathbb{R}$ of transition-independent zs-SG $\mathcal{M}'$, as defined by Bellman's optimality equations in Theorem 3, is a linear map over informed occupancy states. Specifically, if $x \in \mathcal{F}$, then $v_*(x) = 0$; otherwise,*

$$v_*(x) = \sum_{h_{pub}\in\mathcal{H}_{pub}} \mathrm{Pr}(h_{pub}|x) \max_{d_{1,h_{pub}}\in\mathcal{D}_{1,h_{pub}}} \min_{d_{2,h_{pub}}\in\mathcal{D}_{2,h_{pub}}} q_*(o_{(x,h_{pub})}, d_{1,h_{pub}}, d_{2,h_{pub}})$$
$$q_*(o_{(x,h_{pub})}, d_{1,h_{pub}}, d_{2,h_{pub}}) = \rho(o_{(x,h_{pub})}, d_{1,h_{pub}}, d_{2,h_{pub}}) + \gamma v_*(\tau(o_{(x,h_{pub})}, d_{1,h_{pub}}, d_{2,h_{pub}})),$$

*where $o_{(x,h_{pub})}$ denotes the informed occupancy state induced by $(x, h_{pub})$.*

*Proof.* The result follows from Theorem 3, leveraging the linearity of $\rho$ and $\tau$, and the fact that uninformed occupancy states are convex combinations of informed occupancy states.

**Linearity of $\rho$.**   Suppose $x = \sum_{h_{pub}\in\mathcal{H}_{pub}} \mathrm{Pr}(h_{pub}|x) \cdot (o_{(x,h_{pub})} \otimes e_{h_{pub}})$. Then,

$$\rho(x, d_1, d_2) \doteq \sum_s \sum_{h_1,h_2} \sum_{h_{pub}} x(s, h_1, h_2, h_{pub}) \sum_{a_1,a_2} d_1(a_1|h_1, h_{pub}) d_2(a_2|h_2, h_{pub}) r(s, a_1, a_2)$$
$$= \sum_{h_{pub}} \mathrm{Pr}(h_{pub}|x) \cdot \rho(o_{(x,h_{pub})}, d_{1,h_{pub}}, d_{2,h_{pub}}),$$

where $d_{i,h_{pub}}(a_i|h_i) = d_i(a_i|h_i, h_{pub})$.

**Linearity of $\tau$.**   Let $x\prime = \tau(x, d_1, d_2)$, and use the same decomposition of $x$ as above. Then:

$$x\prime(s\prime, (h_i, a_i, z_i)_i, (h_{pub}, w)) \doteq \sum_s x(s, h_1, h_2, h_{pub}) p(s\prime, z_1, z_2, w|s, a_1, a_2) \prod_i d_i(a_i|h_i, h_{pub})$$
$$= \mathrm{Pr}(h_{pub}|x) \cdot \tau(o_{(x,h_{pub})}, d_{1,h_{pub}}, d_{2,h_{pub}})(s\prime, (h_i, a_i, z_i)_i, w).$$

Hence, $\tau(x, d_1, d_2)$ is a convex combination of next informed states.

**Linearity of $v_*$.** From Theorem 3, we have:

$$v_*(x) = \max_{d_1} \min_{d_2} \left[ \rho(x, d_1, d_2) + \gamma v_*(\tau(x, d_1, d_2)) \right]$$

$$= \sum_{h_{\text{pub}}} \Pr(h_{\text{pub}}|x) \cdot \max_{d_{1,h_{\text{pub}}}} \min_{d_{2,h_{\text{pub}}}} \left[ \rho(o_{(x,h_{\text{pub}})}, d_{1,h_{\text{pub}}}, d_{2,h_{\text{pub}}}) + \gamma v_*(\tau(o_{(x,h_{\text{pub}})}, d_{1,h_{\text{pub}}}, d_{2,h_{\text{pub}}})) \right]$$

$$= \sum_{h_{\text{pub}}} \Pr(h_{\text{pub}}|x) \cdot v_*(o_{(x,h_{\text{pub}})}),$$

where the final equality holds by definition of the informed value $v_*(o_{(x,h_{\text{pub}})})$, by an abuse of notation, completing the proof. □

## C.4 Proof of Theorem 5

**Theorem 5.** *The optimal state-value function $v_* : \mathcal{X} \to \mathbb{R}$ is uniformly continuous across uninformed occupancy states. There exists a collection $\bar{\Gamma}_1$ of finite sets $\bar{\Gamma}_2$ of functions $\alpha_2$, each linear over marginal occupancy states $c_2$, such that for any uninformed occupancy state $x$, we have:*

$$v_*(x) = \sum_{h_{pub} \in \mathcal{H}_{pub}} \Pr(h_{pub}|x) \left[ \max_{\bar{\Gamma}_2 \in \bar{\Gamma}_1} \sum_{h_2 \in \mathcal{H}_2} \Pr(h_2|h_{pub}, x) \min_{\alpha_2 \in \bar{\Gamma}_2} \alpha_2(c_{2,(x,h_{pub},h_2)}) \right].$$

*Proof.* The result follows from the definition of the optimal value function of the focal planner and the convex decomposition of uninformed occupancy states $x$ at stage $t$:

$$v_*(x) = \max_{\pi_{1,\ell-t} \in \Pi_{1,\ell-t}} \min_{\pi_{2,\ell-t} \in \Pi_{2,\ell-t}} v_{\pi_{1,\ell-t}, \pi_{2,\ell-t}}(x) \tag{7}$$

$$= \sum_{h_{\text{pub}}} \Pr(h_{\text{pub}}|x) \max_{\pi_{1,\ell-t}} \min_{\pi_{2,\ell-t}} v_{\pi_{1,\ell-t}, \pi_{2,\ell-t}}(o_{(x,h_{\text{pub}})}) \tag{8}$$

$$= \sum_{h_{\text{pub}}} \Pr(h_{\text{pub}}|x) \max_{\pi_{1,\ell-t}} \sum_{h_2} \Pr(h_2|x, h_{\text{pub}}) \min_{\alpha_2 \in \bar{\Gamma}_{2,\pi_{1,\ell-t}}} \alpha_2(c_{2,(x,h_{\text{pub}},h_2)}) \tag{9}$$

$$= \sum_{h_{\text{pub}}} \Pr(h_{\text{pub}}|x) \max_{\bar{\Gamma}_2 \in \bar{\Gamma}_1} \sum_{h_2} \Pr(h_2|x, h_{\text{pub}}) \min_{\alpha_2 \in \bar{\Gamma}_2} \alpha_2(c_{2,(x,h_{\text{pub}},h_2)}). \tag{10}$$

Equation (7) follows from the definition of the optimal value rooted at $x$. Equation (8) uses the convex decomposition of $x$ across informed occupancy states, that is, $x = \sum_{h_{\text{pub}}} \Pr(h_{\text{pub}}|x) \cdot (o_{(x,h_{\text{pub}})} \otimes e_{h_{\text{pub}}})$. Equation (9) follows by expressing $x$ as a convex combination over marginal occupancy states indexed by public and private histories. Equation (10) introduces a collection $\bar{\Gamma}_1$ of sets $\bar{\Gamma}_{2,\pi_{1,\ell-t}}$, one for each focal policy $\pi_{1,\ell-t}$. Although the space of such policies is uncountably infinite—since policies lie in a continuum—each set $\bar{\Gamma}_{2,\pi_{1,\ell-t}}$ in Equation (9) is finite, as it contains only deterministic policy trees $\delta_{2,\ell-t} \in \Delta_{2,\ell-t}$. Each element $\alpha_2 \in \bar{\Gamma}_{2,\pi_{1,\ell-t}}$ is a linear function over marginal occupancy states, defined as $\alpha_2 : (s, h_1) \mapsto v_{\pi_{1,\ell-t}, \delta_{2,\ell-t}}(s, h_1)$—we draw inspiration from the literature on partially observable Markov decision processes [Pineau et al., 2003]. □

# D $\varepsilon$-Optimally Solving $\mathcal{M}$ as $\mathcal{M}'$ via Point-Based Value Iteration

**Existing uniform continuity properties are weaker.** Recent work has established various uniform continuity properties of optimal value functions to support the design of efficient point-based operators [Wiggers et al., 2016, Delage et al., 2023, Cunha et al., 2023]. To formulate these properties precisely, we introduce two notions associated with an uninformed occupancy state: *marginals* and *conditionals*. For any uninformed occupancy state $x$, the *marginal $m_2$* of player 2 is defined as the marginal distribution of $x$ over private histories $h_2 \in \mathcal{H}_2$ and public histories $h_{pub} \in \mathcal{H}_{pub}$:

$$m_2(h_2, h_{pub}) = \sum_{s \in \mathcal{S}} \sum_{h_1 \in \mathcal{H}_1} x(s, (h_1, h_2, h_{pub})).$$

Moreover, for any $x$, and any pair $(h_2, h_{pub})$, the *conditional occupancy state* $c_{2,(x,h_2,h_{pub})}$ is the marginal[1] distribution over $(s, h_1)$ given $(h_2, h_{pub})$, such that:

$$c_{2,(x,h_2,h_{pub})}(s, h_1) = \frac{x(s, (h_1, h_2, h_{pub}))}{m_2(h_2, h_{pub})}.$$

We write $c_2$ to denote the family of such conditionals: $\{c_{2,(x,h_2,h_{pub})} \mid h_2 \in \mathcal{H}_2, \ h_{pub} \in \mathcal{H}_{pub}\}$, and use $c_2 \odot m_2$ to denote a unique uninformed occupancy state $x$ reconstructed from this decomposition:

$$x(s, (h_1, h_2, h_{pub})) = c_{2,(x,h_2,h_{pub})}(s, h_1) \cdot m_2(h_2, h_{pub}),$$

---

[1]Strictly speaking, we should have referred to *marginal occupancy states* as **conditional occupancy states**, in line with their formal definition. Likewise, the *marginal planner* would be more appropriately named the **conditional planner**. We will revise this terminology in the final version of the paper.

for all $s \in \mathcal{S}$ and $(h_1, h_2, h_{pub}) \in \mathcal{H}$. We are now ready to formally present the known uniform continuity properties.

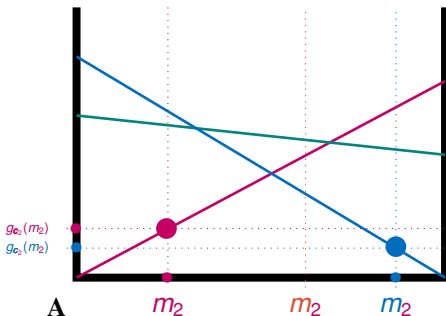 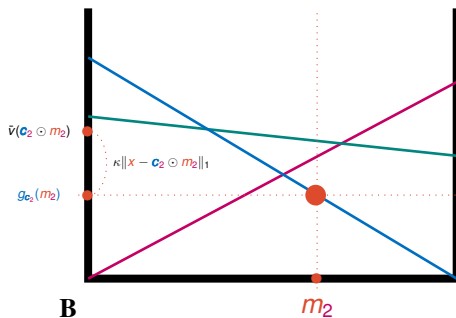

Figure 5: Generalization across marginals of the value function given by a collection $G = \{g_{\boldsymbol{c}_2}, g_{\boldsymbol{c}_2}, g_{\boldsymbol{c}_2}\}$ of linear functions over unknown marginals. Figure **A** shows no generalization on marginal $m_2$ because $m_2 \notin \{m_2, m_2, m_2\}$, *cf.* Theorem D.1. Figure **B** shows generalization over unknown marginal occupancy state $m_2$ from known marginal $m_2$ with offset $\kappa\|x - \boldsymbol{c}_2 \odot m_2\|_1$, *cf.* Theorem D.2. **Best viewed in color**.

**Theorem D.1** (Adapted from Wiggers et al. [2016]). *For any arbitrary $\mathcal{M}'$, the optimal value functions $v_*$ defined in Theorem 4 are convex over marginals, conditioned on a fixed conditional family. That is, there exists a collection $G$ of linear functions over marginals such that for any stage $t$ and any uninformed occupancy state $x = \boldsymbol{c}_2 \odot m_2$, $v_*(x) = \max_{g_{\boldsymbol{c}_2} \in G} g_{\boldsymbol{c}_2}(m_2)$, where each $g_{\boldsymbol{c}_2} : \mathcal{H}_2 \times \mathcal{H}_{pub} \to \mathbb{R}$ is associated with the conditional family $\boldsymbol{c}_2$.*

Wiggers et al. [2016] provides a detailed proof of Theorem D.1, showing that if two uninformed occupancy states share the same conditional family, value generalisation from one to the other is possible. However, this conditional uniform continuity property does not support generalisation across uninformed occupancy states with differing conditionals. Figure 5 (**A**) visualises this limitation. To address this, Delage et al. [2023] combine the conditional property with Lipschitz continuity, thereby enabling generalisation to previously unseen uninformed occupancy states.

**Theorem D.2** (Adapted from Delage et al. [2023]). *For any arbitrary $\mathcal{M}'$, the optimal value functions $v_*$ defined in Theorem 4 are Lipschitz continuous over uninformed occupancy states. That is, there exists a collection $G$ of linear functions over marginals such that, for any stage and any uninformed occupancy state $x = \bar{\boldsymbol{c}}_2 \odot m_2$, $v_*(x) \leq g_{\boldsymbol{c}_2}(m_2) + \kappa\|x - \boldsymbol{c}_2 \odot m_2\|_1$, where $\kappa$ is the Lipschitz constant associated with $v_*$, and $g_{\boldsymbol{c}_2} \in G$ is any function associated with the conditional family $\boldsymbol{c}_2$.*

Despite enabling broader generalisation, Theorem D.2 suffers from loose approximations due to the use of global Lipschitz constants—see Figure 5 (**B**). Furthermore, applying greedy-action selection operators with non-linear value approximations requires evaluating exponentially many decision rules for player 2: $(v, x) \mapsto \operatorname{argmax}_{d_1 \in \mathcal{D}_1} \min_{d_2 \in \mathcal{D}_2} \rho(x, d_1, d_2) + \gamma v(\tau(x, d_1, d_2))$. Delage et al. [2023] implement such operators using linear programs with exponentially many constraints. When $v$ is known, a corresponding linear program takes the form:

$$\max\left\{\upsilon \mid \upsilon \leq \rho(x, d_1, d_2) + \gamma v(\tau(x, d_1, d_2)), \ \forall d_2 \in \mathcal{D}_2\right\},$$

for each $d_1 \in \mathcal{D}_1$, involving $O(|\mathcal{A}_2|^{|\mathcal{H}_2(x) \times \mathcal{H}_{pub}(x)|})$ constraints, where

$$\mathcal{H}_2(x) \times \mathcal{H}_{pub}(x) = \left\{(h_2, h_{pub}) \in \mathcal{H}_2 \times \mathcal{H}_{pub} \mid \Pr(h_2, h_{pub} \mid x) > 0\right\}.$$

To mitigate this burden, Delage et al. [2023] restrict attention to previously encountered (stochastic) decision rules rather than the full decision space. Nevertheless, these limitations hinder algorithmic efficiency and highlight the need for alternative approaches. Our uniform continuity property, presented in Theorem 5, is strictly stronger than all previously established results, enabling seamless generalisation across arbitrary uninformed occupancy states. Notably, prior work typically defined the optimal value function over marginal distributions, whereas we define it over conditional distributions of uninformed occupancy states—following the approach commonly adopted in partially observable Markov decision processes [Smallwood and Sondik, 1973, Sondik, 1978] and decentralised variants [Dibangoye et al., 2016]. This shift unveils markedly stronger uniform continuity properties.

## D.1 Proof of Corollary 1

**Corollary 1.** *The optimal action-value function $q_* : \mathcal{X} \times \mathcal{D}_1 \to \mathbb{R}$ is uniformly continuous across uninformed occupancy states. There exists a collection $\Phi_1$ of finite sets $\Phi_2$ of functions $\phi_2$, each linear over marginal occupancy states $c_2$ and private decision rules $d_1$. Thus, for any uninformed occupancy state $x$ and private decision rule $d_1$,*

$$q_*(x, d_1) = \sum_{h_{pub}} \Pr(h_{pub}|x) \max_{\phi_1 \in \Delta(\Phi_1)} \sum_{\Phi_2 \in \Phi_1} \phi_1(\Phi_2) \sum_{h_2} \Pr(h_2|x, h_{pub}) \min_{a_2, \phi_2 \in \Phi_2} \phi_2(c_{2,(x,h_{pub},h_2)}, d_1, a_2),$$

*Proof.* The result follows from the uniform continuity of $v_*$ over uninformed occupancy states (*cf.* Theorem 5) and the convex decomposition of such states. Starting from the definition of the optimal action-value function: for any uninformed occupancy state $x$ and decision rule $d_1$,

$$q_*(x, d_1) \doteq \min_{d_2 \in \mathcal{D}_2} \max_{\pi_{1,\ell-t} \in \Pi_{1,\ell-t}} \min_{\pi_{2,\ell-t} \in \Pi_{2,\ell-t}} q_{\pi_{1,\ell-t}, \pi_{2,\ell-t}}(x, d_1, d_2).$$

Expanding the occupancy state over public observation histories $h_{\text{pub}} \in \mathcal{H}_{\text{pub}}$ yields:

$$q_*(x, d_1) = \min_{d_2} \sum_{h_{\text{pub}}} \Pr(h_{\text{pub}}|x) \max_{\pi_{1,\ell-t}} \min_{\pi_{2,\ell-t}} q_{\pi_{1,\ell-t}, \pi_{2,\ell-t}}(o_{(x,h_{\text{pub}})}, d_1, d_2).$$

Refining $o_{(x,h_{\text{pub}})}$ over private histories $h_2 \in \mathcal{H}_2$ of player 2, and defining finite set $\Phi_{2,\pi_{1,\ell-t}} = \{q_{\pi_{1,\ell-t}, \delta_{2,\ell-t}} : \delta_{2,\ell-t} \in \Delta_{2,\ell-t}\}$ of function $\phi_2$ linear across marginal occupancy states and decision rules of player 1, induced by policy trees $\delta_{2,\ell-t} \in \Delta_{2,\ell-t}$ of player 2, we obtain:

$$= \min_{d_2} \sum_{h_{\text{pub}}} \Pr(h_{\text{pub}}|x) \max_{\pi_{1,\ell-t}} \sum_{h_2} \Pr(h_2|x, h_{\text{pub}}) \min_{\phi_2 \in \Phi_{2,\pi_{1,\ell-t}}} \sum_{a_2} d_{2,h_{\text{pub}}}(a_2|h_2) \cdot \phi_2(c_{2,(x,h_{\text{pub}},h_2)}, d_1, a_2).$$

Letting $\Phi_1 = \{\Phi_{2,\pi_{1,\ell-t}} | \pi_{1,\ell-t} \in \Pi_{1,\ell-t}\}$, and observing that $d_{2,h_{\text{pub}}}(a_2|h_2) = d_2(a_2|h_2, h_{\text{pub}})$, we can apply Neumann [1928] to exchange the min–max ordering:

$$q_*(x, d_1) = \sum_{h_{\text{pub}}} \Pr(h_{\text{pub}}|x) \max_{\phi_1 \in \Delta(\Phi_1)} \sum_{\Phi_2 \in \Phi_1} \phi_1(\Phi_2) \sum_{h_2} \Pr(h_2|x, h_{\text{pub}}) \min_{a_2, \phi_2 \in \Phi_2} \phi_2(c_{2,(x,h_{\text{pub}},h_2)}, d_1, a_2),$$

which completes the proof. $\square$

## D.2 Proof of Theorem 6

**Theorem 6.** *Let $o$ be an informed occupancy state. Then the decision rule $d_1$ maximising $q(o, \cdot)$ can be computed as the solution of the following linear program with:*

- *$O(|\Phi_1| \cdot |\mathcal{H}_1(o)| \cdot |\mathcal{A}_1|)$ **variables**,*
- *$O(|\Phi_1| \cdot |\Phi_2^*| \cdot |\mathcal{H}_2(o)| \cdot |\mathcal{A}_2|)$ **constraints**,*

*where $\Phi_2^*$ denotes the largest set of linear functions within any $\Phi_2 \in \Phi_1$. The linear program is:*

*Maximise*    $\sum_{\Phi_2 \in \Phi_1} \sum_{h_2 \in \mathcal{H}_2(o)} \Pr(h_2|o) \cdot \upsilon(h_2, \Phi_2)$

*Subject to*    $\sum_{a_1 \in \mathcal{A}_1} \sum_{\Phi_2 \in \Phi_1} \xi_1(a_1, \Phi_2|h_1) = 1, \quad \forall h_1 \in \mathcal{H}_1(o),$

              $\upsilon(h_2, \Phi_2) \leq \sum_{h_1} \sum_{a_1} \xi_1(a_1, \Phi_2|h_1) \sum_{s \in S} \phi_2(s, h_1, a_1, a_2) \cdot c_{2,(o,h_2)}(s, h_1),$

              $\forall \Phi_2, \forall \phi_2 \in \Phi_2, \forall a_2 \in \mathcal{A}_2, \forall h_2 \in \mathcal{H}_2(o),$

*where $\mathcal{H}_i(o)$ denotes the finite set of private histories of player $i$ reachable in $o$. The variable $\xi_1(a_1, \Phi_2|h_1)$ encodes the probability of taking action $a_1$ in history $h_1$, assuming the value model $\Phi_2$ is drawn from $\phi_1$. The inner constraint ensures that the worst-case evaluation $\upsilon(h_2, \Phi_2)$ is always pessimistic—i.e., no matter how the opponent reacts, the value function bound holds.*

*Proof.* Corollary 1 shows that $q_*(o, d_1)$ is the maximum over concave combinations of linear functions $\phi_2$, each defined over marginal occupancy states, *i.e.*,

$$q_*(o, d_1) = \max_{\phi_1 \in \Delta(\Phi_1)} \sum_{\Phi_2 \in \Phi_1} \phi_1(\Phi_2) \sum_{h_2} \Pr(h_2|o) \min_{a_2 \in \mathcal{A}_2, \phi_2 \in \Phi_2} \phi_2(c_{2,(o,h_2)}, d_1, a_2).$$

If we let $\xi_1(a_1, \Phi_2|h_1) = \phi_1(\Phi_2) \cdot d_1(a_1|h_1)$ encode the probability of taking action $a_1$ in history $h_1$, assuming the value model is drawn from $\phi_1$, then:

$$\xi_1^* \in \text{argmax}_{\xi_1 \in \Delta(\Phi_1) \times \mathcal{D}_1} \sum_{\Phi_2 \in \Phi_1} \sum_{h_2} \min_{a_2 \in \mathcal{A}_2, \phi_2 \in \Phi_2} \text{Pr}(h_2|o) \cdot \phi_1(\Phi_2) \cdot \phi_2(c_{2,(o,h_2)}, d_1, a_2).$$

To extract a greedy rule, we represent $q_*(o, d_1)$ as a linear objective with auxiliary variables $\upsilon(h_2, \Phi_2)$ that lower-bound the worst-case value against each opponent history and linear function, *i.e.,*

$$\upsilon(h_2, \Phi_2) \leq \phi_1(\Phi_2) \cdot \phi_2(c_{2,(o,h_2)}, d_1, a_2), \quad \forall \Phi_2, \forall \phi_2 \in \Phi_2, \forall a_2 \in \mathcal{A}_2, \forall h_2 \in \mathcal{H}_2(o).$$

The linearity in $d_1$ and marginal state structure ensures the objective and constraints remain linear,

$$\upsilon(h_2, \Phi_2) \leq \sum_{h_1} \sum_{a_1} \xi_1(a_1, \Phi_2|h_1) \sum_{s \in \mathcal{S}} \phi_2(s, h_1, a_1, a_2) \cdot c_{2,(o,h_2)}(s, h_1).$$

This yields a valid linear program whose optimum corresponds to the desired decision rule. $\quad\square$

It is worth noting that each set $\Phi_2$ is derived from a corresponding set $\Gamma_2$; that is, for every $\phi_2^\nu \in \Phi_2$, there exists a mapping $\nu \colon \mathcal{Z}_2 \times \mathcal{W} \mapsto \Gamma_2$ such that:

$$\phi_2^\nu(s, h_1, a_1, a_2) = r(s, a_1, a_2) + \gamma \sum_{s', z_1, z_2, w} p(s', z_1, z_2, w|s, a_1, a_2) \cdot \nu(z_2, w)(s', (h_1, z_1, a_1)).$$

Evidently, $|\Phi_2|$ is exponential in the worst case, i.e., $|\Phi_2| \in O(|\Gamma_2|^{|\mathcal{Z}_2| \cdot |\mathcal{W}|})$. To enhance scalability when selecting greedy decision rules, one may instead optimise directly over $\Gamma_1$, thereby reducing the number of constraints from exponential to linear.

**Theorem D.3.** *Let $o$ be an informed occupancy state. Then the decision rule $d_1$ that maximises $q(o, \cdot)$ can be obtained as the solution to the following linear program:*

- $O(|\Gamma_1| \cdot |\mathcal{H}_2(o)| \cdot |\mathcal{A}_2| \cdot |\mathcal{Z}_2| \cdot |\mathcal{W}|)$ **variables***,*

- $O(|\Gamma_1| \cdot |\Gamma_2^*| \cdot |\mathcal{H}_2(o)| \cdot |\mathcal{A}_2| \cdot |\mathcal{Z}_2| \cdot |\mathcal{W}|)$ ***constraints***,*

*where $\Gamma_2^*$ denotes the largest value function set across all $\Gamma_2 \in \Gamma_1$. The linear program is:*

*Maximise* $\quad \displaystyle\sum_{h_2 \in \mathcal{H}_2(o)} f_\theta(h_2)$

*Subject to* $\quad \displaystyle\sum_{\Gamma_2 \in \Gamma_1} \sum_{a_1 \in \mathcal{A}_1} \theta_{\Gamma_2}(a_1|h_1) = 1, \quad \forall h_1 \in \mathcal{H}_1(o)$

$$f_\theta(h_2) \leq \sum_{\Gamma_2 \in \Gamma_1} \sum_{z_2 \in \mathcal{Z}_2} \sum_{w \in \mathcal{W}} \beta_{\Gamma_2}(h_2, a_2, z_2, w), \quad \forall h_2 \in \mathcal{H}_2(o), \forall a_2 \in \mathcal{A}_2$$

$$\beta_{\Gamma_2}(h_2, a_2, z_2, w) \leq \sum_{a_1 \in \mathcal{A}_1} \sum_{h_1 \in \mathcal{H}_1} \theta_{\Gamma_2}(a_1|h_1) \cdot g_{\Gamma_2, \alpha_2}(h_1, h_2, a_1, a_2, z_2, w),$$

$$\forall \Gamma_2 \in \Gamma_1, \forall \alpha_2 \in \Gamma_2, \forall h_2 \in \mathcal{H}_2(o), \forall a_2 \in \mathcal{A}_2, \forall z_2 \in \mathcal{Z}_2, \forall w \in \mathcal{W}$$

*where $\mathcal{H}_i(o)$ is the finite set of private histories of player $i$ reachable in $o$.*

*The variable $\theta_{\Gamma_2}(a_1|h_1)$ encodes the probability of taking action $a_1$ at history $h_1$, under value model $\Gamma_2$. The inner constraint ensures that $f_\theta(h_2)$ is a pessimistic estimate—i.e., it remains valid regardless of how the opponent responds. This is achieved by ensuring the intermediate evaluation $\beta_{\Gamma_2}(\cdot)$ is also pessimistic—i.e., valid for all $\alpha_2 \in \Gamma_2$. The value of following policy $\alpha_2$ is given by:*

$$g_{\Gamma_2, \alpha_2} \colon (h_1, h_2, a_1, a_2, z_2, w) \mapsto \sum_{s \in \mathcal{S}} o(s, h_1, h_2) \cdot \beta_2(s, h_1, a_1, a_2, z_2, w),$$

$$\beta_2 \colon (s, h_1, a_1, a_2, z_2, w) \mapsto r(s, a_1, a_2) + \gamma \sum_{s' \in \mathcal{S}} \sum_{z_1 \in \mathcal{Z}_1} p(s', z_1, z_2, w|s, a_1, a_2) \cdot \alpha_2(s', h_1, a_1, z_1).$$

*Proof.* The proof starts with the definition of the greedy decision rule selection at informed occupancy state $o$ at stage $t$, assuming uniformly continuous value function $v$. Let $q \colon (o, d_1, d_2) \mapsto \rho(o, d_1, d_2) + \gamma v(\tau(o, d_1, d_2))$. Then, $d_{1,o} \in \text{argmax}_{d_1} \min_{d_2} q(o, d_1, d_2)$. The following holds by the application of the uniform continuity property of the optimal value function from Theorem 5:

$$v(\tau(o, d_1, d_2)) = \max_{\Gamma_2 \in \Gamma_1} \sum_{h_2, z_2, a_2, w} \text{Pr}(h_2, a_2, z_2, w|\tau(o, d_1, d_2)) \min_{\alpha_2 \in \Gamma_2} \alpha_2(c_{2,(\tau(o,d_1,d_2)),(h_2,a_2,z_2,w)}).$$

If we replace the $\max_{\Gamma_2 \in \Gamma_1}$ by $\max_{\xi \in \Delta(\Gamma_1)}$ then there is no loss in optimality, *i.e.,*

$$= \max_{\xi \in \Delta(\Gamma_1)} \sum_{h_2, z_2, a_2, w} \sum_{\Gamma_2 \in \Gamma_1} \min_{\alpha_2 \in \Gamma_2} \xi(\Gamma_2) \cdot \Pr(h_2, a_2, z_2, w | \tau(o, d_1, d_2)) \alpha_2(c_{2,(\tau(o,d_1,d_2),(h_2,a_2,z_2,w))}).$$

Notice that the product rule provides us with the following relation:

$$\Pr(s\prime, h_1, a_1, z_1, h_2, a_2, z_2, w | o, d_1, d_2)$$
$$= \Pr(h_2, a_2, z_2, w | \tau(o, d_1, d_2)) \cdot c_{2,(\tau(o,d_1,d_2),(h_2,a_2,z_2,w))}(s\prime, h_1, a_1, z_1)$$
$$= d_1(a_1 | h_1) \cdot d_2(a_2 | h_2) \cdot \sum_s o(s, h_1, h_2) \cdot p(s\prime, z_1, z_2, w | s, a_1, a_2).$$

Exploiting this insight along with the linearity of $\alpha_2$ yields:

$$v(\tau(o, d_1, d_2)) = \max_{\xi \in \Delta(\Gamma_1)} \sum_{h_2, z_2, a_2, w} \sum_{\Gamma_2 \in \Gamma_1} \min_{\alpha_2 \in \Gamma_2}$$
$$\sum_{s, s\prime, h_1, a_1, z_1} \xi(\Gamma_2) \cdot \alpha_2(s\prime, h_1, a_1, z_1) \cdot d_1(a_1 | h_1) \cdot d_2(a_2 | h_2) \cdot$$
$$o(s, h_1, h_2) \cdot p(s\prime, z_1, z_2, w | s, a_1, a_2).$$

Define the following two intermediate functions $g_{\Gamma_2, \alpha_2}$ and $\beta_2$

$$g_{\Gamma_2, \alpha_2} : (h_1, h_2, a_1, a_2, z_2, w) \mapsto \sum_{s \in \mathcal{S}} o(s, h_1, h_2) \cdot \beta_2(s, h_1, a_1, a_2, z_2, w)$$
$$\beta_2(s, h_1, a_1, a_2, z_2, w) \doteq r(s, a_1, a_2) + \gamma \sum_{s\prime \in \mathcal{S}} \sum_{z_1 \in \mathcal{Z}_1} p(s\prime, z_1, z_2, w | s, a_1, a_2) \cdot \alpha_2(s\prime, h_1, a_1, z_1).$$

Consequently, the action value can be rewritten as follows:

$$q(o, d_1, d_2) = \max_{\xi \in \Delta(\Gamma_1)} \sum_{h_2, a_2} d_2(a_2 | h_2) \sum_{\Gamma_2 \in \Gamma_1} \xi(\Gamma_2)$$
$$\sum_{z_2, w} \min_{\alpha_2 \in \Gamma_2} \sum_{h_1, a_1} d_1(a_1 | h_1) \cdot g_{\Gamma_2, \alpha_2}(h_1, h_2, a_1, a_2, z_2, w).$$

Let us define the decision variable $\theta_{\Gamma_2}(a_1 | h_1) \doteq d_1(a_1 | h_1) \cdot \xi(\Gamma_2)$ then our greedy decision rule is the solution of the following maximin optimisation problem:

$$\max_\theta \min_{d_2} \sum_{h_2, a_2} d_2(a_2 | h_2) \sum_{\Gamma_2 \in \Gamma_1} \sum_{z_2, w} \min_{\alpha_2 \in \Gamma_2} \sum_{h_1, a_1} \theta_{\Gamma_2}(a_1 | h_1) \cdot g_{\Gamma_2, \alpha_2}(h_1, h_2, a_1, a_2, z_2, w).$$

Using Wald's maximin model we can convert this maximin optimisation problem into a maximisation mathematical program, *i.e.,*

| | |
|---|---|
| Maximise | $\sum_{h_2 \in \mathcal{H}_2(o)} f_\theta(h_2)$ |
| Subject to | $\sum_{\Gamma_2 \in \Gamma_1} \sum_{a_1 \in \mathcal{A}_1} \theta_{\Gamma_2}(a_1 | h_1) = 1, \quad \forall h_1 \in \mathcal{H}_1(o)$ |
| | $f_\theta(h_2) \leq \sum_{\Gamma_2 \in \Gamma_1} \sum_{z_2 \in \mathcal{Z}_2} \sum_{w \in \mathcal{W}} \beta_{\Gamma_2}(h_2, a_2, z_2, w), \quad \forall h_2 \in \mathcal{H}_2(o), \forall a_2 \in \mathcal{A}_2$ |
| | $\beta_{\Gamma_2}(h_2, a_2, z_2, w) \leq \sum_{a_1 \in \mathcal{A}_1} \sum_{h_1 \in \mathcal{H}_1} \theta_{\Gamma_2}(a_1 | h_1) \cdot g_{\Gamma_2, \alpha_2}(h_1, h_2, a_1, a_2, z_2, w),$ |
| | $\forall \Gamma_2 \in \Gamma_1, \forall \alpha_2 \in \Gamma_2, \forall h_2 \in \mathcal{H}_2(o), \forall a_2 \in \mathcal{A}_2, \forall z_2 \in \mathcal{Z}_2, \forall w \in \mathcal{W}$ |

Then, the solutin of the linear program in the theorem is the greedy decision rule of the focal player, which ends the proof. □

### D.3 Proof of Corollary 2

**Corollary 2.** *Let $v$ and $q$ be the current state- and action-value functions represented by finite collections $\Gamma_1$ of sets $\Gamma_2$, and $\Phi_1$ of sets $\Phi_2$, respectively. Let $o$ be an informed occupancy state, and let $\xi_1$ denote the solution of the greedy linear program from Theorem 6 at $o$. We define an updated value function $v\prime$ by augmenting $\Gamma_1$ with a new set $\Gamma_{2,(\mathcal{C}_2\prime, \xi_1)}$ of linear functions $\alpha_{2,(c_2)}$ given by:*

$$\alpha_{2,(c_2)} = \sum_{\Phi_2 \in \Phi_1} \operatorname{argmin}_{\alpha_2^{\phi_2, a_2} : \phi_2 \in \Phi_2, a_2 \in \mathcal{A}_2} \alpha_2^{\phi_2, a_2}(c_2)$$
$$\alpha_2^{\phi_2, a_2}(s, h_1) = \sum_{a_1} \xi_1(a_1, \Phi_2 | h_1) \cdot \phi_2(s, h_1, a_1, a_2).$$

*Then $v\prime(x) \geq v(x)$ for any uninformed occupancy state $x$ induced by $\mathcal{C}_2\prime$, and $v\prime(x) > v(x)$ for at least one such $x$ if the greedy update yields a strict improvement.*

*Proof.* We are given that the value function $v$ is represented by a collection $\Gamma_1$ of finite sets $\Gamma_2$, where each $\Gamma_2$ contains functions linear over marginal occupancy states. Let $o$ be an informed occupancy state and $\xi_1$ the greedy decision rule obtained by solving the linear program in Theorem 6 at $o$.

We define a new set of linear functions $\Gamma_{2,(\mathcal{C}_2',\xi_1)}$ supported on the sampled marginal states $\mathcal{C}_2'$. For each $c_2 \in \mathcal{C}_2'$, define the linear function

$$\alpha_{2,(c_2)} = \sum_{\Phi_2 \in \Phi_1} \text{argmin}_{\alpha_2^{\phi_2,a_2}:\ \phi_2 \in \Phi_2,\ a_2 \in \mathcal{A}_2}\ \alpha_2^{\phi_2,a_2}(c_2)$$

$$\alpha_2^{\phi_2,a_2}(s,h_1) = \sum_{a_1} \xi_1(a_1,\Phi_2|h_1) \cdot \phi_2(s,h_1,a_1,a_2).$$

These functions satisfy the constraints of the linear program at $o$ and define a new set $\Gamma_{2,(\mathcal{C}_2',\xi_1)}$. We then update the value function by setting

$$\Gamma_1' = \Gamma_1 \cup \left\{ \Gamma_{2,(\mathcal{C}_2',\xi_1)} \right\}.$$

Let $v\prime$ be the value function induced by $\Gamma_1'$, and fix an uninformed occupancy state $x$ such that all the marginal states $c_{2,(x,h_{\text{pub}},h_2)}$ involved in its convex decomposition lie in $\mathcal{C}_2'$. Then, by construction of improved state-value function $v\prime$,

$$v\prime(x) = \max_{\Gamma_2 \in \Gamma_1'} \sum_{h_{\text{pub}}} \Pr(h_{\text{pub}}|x) \sum_{h_2} \Pr(h_2|h_{\text{pub}},x) \min_{\alpha_2 \in \Gamma_2} \alpha_2(c_{2,(x,h_{\text{pub}},h_2)}).$$

Since we have added a new set of functions that are constructed to satisfy the linear program constraints at $o$, this new maximum is at least as large as before. Thus,

$$v\prime(x) \geq v(x).$$

Finally, if the greedy linear program solution $\xi_1$ at $o$ strictly improves the linear program objective compared to the current value function $v$, then there exists at least one marginal occupancy state $c_2 \in \mathcal{C}_2'$ where the new linear function yields strictly higher value than all previous ones. This yields:

$$v\prime(x) > v(x)$$

for some $x$ whose decomposition includes that $c_2$. □

## D.4 Algorithms

---

**Algorithm 1** PBVI for $\mathcal{M}'$ (resp. $\mathcal{M}$).

```
function PBVI()
Initialise C_{2,0:ℓ}, O_{0:ℓ}.
Initialise Γ_{1,t} ← ∅ for all t ∈ {0, ... , ℓ}.
while has not converged do
   for t = ℓ, ... , 0 do
      improve(Γ_{1,t}).
   end for
   for t = 0, ... , ℓ do
      (C_{2,t}, O_t) ← expand(C_{2,t}, O_t).
   end for
end while
```

```
function improve(Γ_{1,t+1})
for o ∈ O_t do
   ξ_1 ← LP(Γ_{1,t+1}, o)
   Γ_{1,t} ← Γ_{1,t} ∪ {Γ_{2,t,(C_2,ξ_1)}}
end for
```

---

**Algorithm 2** Bounded Pruning.

```
function BoundedPruning(Γ_1, O')
for Γ_2 ∈ Γ_1 do
   refCount(Γ_2) ← 0.
end for
for o ∈ O' do
   Γ_{2,o} ← argmax_{Γ_2 ∈ Γ_1} ∑_{h_2} Pr(h_2|o) min_{α_2 ∈ Γ_2} α_2(c_{2,(o,h_2)})
   refCount(Γ_{2,o}) ← refCount(Γ_{2,o}) + 1
end for
return {Γ_2 ∈ Γ_1 | refCount(Γ_2) > 0}
```

---

**Algorithm 3** Redundant Informed Occupancy State Pruning.

```
function PruneStates(𝒪′, Γ₁, ϵ)
Initialise 𝒪° ← ∅
for o ∈ 𝒪′ do
    Γ₂,ₒ ← argmax_{Γ₂∈Γ₁} ∑_{h₂} Pr(h₂|o) min_{α₂∈Γ₂} α₂(c₂,(o,h₂))
end for
for o ∈ 𝒪′ do
    isRedundant ← false
    for o′ ∈ 𝒪° do
        if | ∑_{h₂} Pr(h₂|o) min_{α₂∈Γ₂,ₒ} α₂(c₂,(o,h₂)) − ∑_{h₂} Pr(h₂|o) min_{α₂∈Γ₂,ₒ′} α₂(c₂,(o,h₂))| ≤ ϵ then
            isRedundant ← true and break
        end if
    end for
    if ¬isRedundant then
        𝒪° ← 𝒪° ∪ {o}
    end if
end for
return 𝒪°
```

## D.5 Proof of Theorem 7

**Theorem 7.** *For any marginal occupancy sample sets $\mathcal{C}\prime_{2,0:\ell}$, the exploitability of the focal policy obtained via PBVI and evaluated at the initial state distribution, is bounded as*

$$\varepsilon \leq \frac{4m\delta}{(1-\gamma)^2} \cdot [1 + (\ell+1)\gamma^{\ell+2} - (\ell+2)\gamma^{\ell+1}].$$

*Proof.* Let $\pi_1$ be an optimal focal policy with value $v_{1,*}(b)$. Let $(x_0, \ldots, x_\ell)$ denote the sequence of uninformed occupancy states induced by $\pi_1$ and the opponent policy $\pi_2$, for which PBVI yields the worst estimate. Let $(x'_0, \ldots, x'_\ell)$ be the closest sequence of uninformed occupancy states to $(x_0, \ldots, x_\ell)$ in $\ell_1$-norm, induced by the sampled marginal set $\mathcal{C}'_{2,0:\ell}$. As a consequence, the following inequality holds $\|x_t - x'_t\|_1 \leq \delta$ for any stage $t$. Let $v_1$ be the approximate value function, and $\pi'_1$ the induced focal policy computed by PBVI over $\mathcal{C}'_{2,0:\ell}$. Let $v_*$ and $v$ be the value functions induced by pairs of behavioural strategies, each linear over uninformed occupancy states, such that $v_*(b) = v_{1,*}(b)$ and $v(b) = v_1(b)$, respectively. These functions always exist for a fixed joint policy, *e.g.*, $v_* = v_{\pi_1,\pi_2}$. Then,

$$\varepsilon \doteq v_{1,*}(b) - \min_{\pi'_2 \in \Pi_2} v_{\pi'_1,\pi'_2}(b)$$
$$= v_{1,*}(b) - v_1(b) + v_1(b) - \min_{\pi'_2 \in \Pi_2} v_{\pi'_1,\pi'_2}(b) \quad \text{(adding zero).}$$

Since the values of the focal and uninformed planners coincide at the initial state distribution, i.e., $v_{1,*}(b) = v_*(x_0)$ and $v_1(b) = v(x_0)$, we have:

$$v_{1,*}(b) - v_1(b) = v_*(x_0) - v(x_0)$$
$$= \left( \sum_{t=0}^{\ell} \gamma^t \cdot \rho(x_t, d_{1,t}, d_{2,t}) \right) - v(x_0) \quad \text{(by definition)}$$
$$= \left( \sum_{t=0}^{\ell} \gamma^t \cdot \rho(x_t, d_{1,t}, d_{2,t}) \right) - \sum_{t=1}^{\ell} \gamma^t(v(x_t) - v(x_t)) - v(x_0) \quad \text{(adding zero).}$$

Using the convention $v_{\ell+1}(\cdot) \doteq 0$, we rearrange terms:

$$= \sum_{t=0}^{\ell} \gamma^t \cdot \rho(x_t, d_{1,t}, d_{2,t}) + \left( \gamma^{\ell+1} v(x_{\ell+1}) + \sum_{t=1}^{\ell} \gamma^t v(x_t) \right) - \left( \gamma^0 v(x_0) + \sum_{t=1}^{\ell} \gamma^t v(x_t) \right)$$
$$= \sum_{t=0}^{\ell} \gamma^t \cdot \rho(x_t, d_{1,t}, d_{2,t}) + \sum_{t=0}^{\ell} \gamma^{t+1} v(x_{t+1}) - \sum_{t=0}^{\ell} \gamma^t v(x_t)$$
$$= \sum_{t=0}^{\ell} \gamma^t \left( \rho(x_t, d_{1,t}, d_{2,t}) + \gamma v(x_{t+1}) - v(x_t) \right)$$
$$= \sum_{t=0}^{\ell} \gamma^t \left( q(x_t, d_{1,t}, d_{2,t}) - v(x_t) \right).$$

Now substitute $x'_t$ in place of $x_t$:

$$= \sum_{t=0}^{\ell} \gamma^t \left( q(x_t, d_{1,t}, d_{2,t}) - q(x_t, d_{1,t}, d_{2,t}) + q(x_t, d_{1,t}, d_{2,t}) - v(x_t) \right)$$

$$= \sum_{t=0}^{\ell} \gamma^t \left( q(x_t, d_{1,t}, d_{2,t}) - q(x'_t, d_{1,t}, d_{2,t}) + q(x'_t, d_{1,t}, d_{2,t}) - v(x_t) \right).$$

Because the greedy rule for $q(x'_t, \cdot, \cdot)$ achieves value $v(x'_t)$, we have:

$$\leq \sum_{t=0}^{\ell} \gamma^t \left( q(x_t, d_{1,t}, d_{2,t}) - q(x'_t, d_{1,t}, d_{2,t}) + v(x'_t) - v(x_t) \right)$$

$$= \sum_{t=0}^{\ell} \gamma^t \left( q(x_t, d_{1,t}, d_{2,t}) - v(x_t) - q(x'_t, d_{1,t}, d_{2,t}) + v(x'_t) \right)$$

$$= \sum_{t=0}^{\ell} \gamma^t \left( q(\cdot, d_{1,t}, d_{2,t}) - v(\cdot) \right) \cdot (x_t - x'_t).$$

Applying Hölder's inequality, using the definition of $\delta$, and the fact that $r$ is bounded:

$$v_{1,*}(b) - v_1(b) \leq \sum_{t=0}^{\ell} \gamma^t \cdot \| q(\cdot, d_{1,t}, d_{2,t}) - v(\cdot) \|_{\infty} \cdot \| x_t - x'_t \|_1$$

$$\leq \delta \sum_{t=0}^{\ell} \gamma^t \cdot \| q(\cdot, d_{1,t}, d_{2,t}) - v(\cdot) \|_{\infty}$$

$$\leq 2m\delta \sum_{t_0=0}^{\ell} \gamma^{t_0} \sum_{t_1=t_0}^{\ell} \gamma^{t_1 - t_0}, \quad (q \text{ and } v \text{ being linear across } x_t)$$

$$= 2m\delta \sum_{t_0=0}^{\ell} \sum_{t_1=t_0}^{\ell} \gamma^{t_1}$$

$$= 2m\delta \sum_{t_0=0}^{\ell} \frac{\gamma^{t_0} - \gamma^{\ell+1}}{1-\gamma}$$

$$= \frac{2m\delta}{1-\gamma} \sum_{t_0=0}^{\ell} (\gamma^{t_0} - \gamma^{\ell+1})$$

$$= \frac{2m\delta}{(1-\gamma)^2} \left[ 1 + (\ell+1)\gamma^{\ell+2} - (\ell+2)\gamma^{\ell+1} \right].$$

A similar argument yields for this part $v_1(b) - \min_{\pi'_2 \in \Pi_2} v_{\pi'_1, \pi'_2}(b)$. Let $\pi_2$ be a best-response to the focal policy $\pi'_1$ induced by $v_1(b)$. Let $v_{\pi'_1, *}$ and $v_{\pi'_1}$ be the value functions induced by the pairs of behavioural strategies, each linear over uninformed occupancy states, such that $v_{\pi'_1, *}(b) = \min_{\pi'_2 \in \Pi_2} v_{\pi'_1, \pi'_2}(b)$ and $v_{\pi'_1}(b) = v_1(b)$, respectively.

$$v_1(b) - \min_{\pi'_2 \in \Pi_2} v_{\pi'_1, \pi'_2}(b) = v_{\pi'_1}(x_0) - v_{\pi'_1, *}(x_0).$$

Let $(x_0, \ldots, x_\ell)$ denote the sequence of uninformed occupancy states induced by $\pi'_1$ and the selected best-response $\pi_2$. Let $x'_0, \ldots, x'_\ell$ be the closezt sequence of uninformed occupancy states to $(x_0, \ldots, x_\ell)$ in $\ell_1$-norm, induced by the sampled marginal set $\mathcal{C}'_{2,0:\ell}$. Then, it follows that:

$$= v_{\pi'_1}(x_0) - \left( \sum_{t=0}^{\ell} \gamma^t \cdot \rho(x_t, d_{1,t}, d_{2,t}) \right), \quad \text{(by Definition)}$$

$$= v_{\pi'_1}(x_0) + \sum_{t=1}^{\ell} \gamma^t \cdot (v_{\pi'_1}(x_t) - v_{\pi'_1}(x_t)) - \left( \sum_{t=0}^{\ell} \gamma^t \cdot \rho(x_t, d_{1,t}, d_{2,t}) \right), \quad \text{(adding zero).}$$

Using the convention $v_{\pi'_1}(x_{\ell+1}) \doteq 0$, we rearrange terms:

$$= (v_{\pi'_1}(x_0) + \sum_{t=1}^{\ell} \gamma^t \cdot v_{\pi'_1}(x_t)) - (\gamma^{\ell+1} \cdot v_{\pi'_1}(x_{\ell+1}) + \sum_{t=1}^{\ell} \gamma^t \cdot v_{\pi'_1}(x_t)) - (\sum_{t=0}^{\ell} \gamma^t \cdot \rho(x_t, d_{1,t}, d_{2,t}))$$

$$= \sum_{t=0}^{\ell} \gamma^t \cdot v_{\pi'_1}(x_t) - \sum_{t=0}^{\ell} \gamma^t \cdot \gamma v_{\pi'_1}(x_{t+1}) - \sum_{t=0}^{\ell} \gamma^t \cdot \rho(x_t, d_{1,t}, d_{2,t})$$

$$= \sum_{t=0}^{\ell} \gamma^t \cdot (v_{\pi'_1}(x_t) - \gamma v_{\pi'_1}(x_{t+1}) - \rho(x_t, d_{1,t}, d_{2,t}))$$

$$= \sum_{t=0}^{\ell} \gamma^t \cdot (v_{\pi'_1}(x_t) - [\rho(x_t, d_{1,t}, d_{2,t}) + \gamma v_{\pi'_1}(x_{t+1})])$$

$$= \sum_{t=0}^{\ell} \gamma^t \cdot (v_{\pi'_1}(x_t) - q_{\pi'_1}(x_t, d_{1,t}, d_{2,t})), \quad \text{(by Definition)}$$

$$= \sum_{t=0}^{\ell} \gamma^t \cdot (v_{\pi'_1}(x_t) - q_{\pi'_1}(x_t, d_{1,t}, d_{2,t}) - q_{\pi'_1}(x'_t, d_{1,t}, d_{2,t}) + q_{\pi'_1}(x'_t, d_{1,t}, d_{2,t})), \quad \text{(adding zero).}$$

Because the greedy rules in $q_{\pi'_1}(x'_t, d_{1,t}, d_{2,t})$ achieves value $v_{\pi'_1}(x'_t)$, we have:

$$\leq \sum_{t=0}^{\ell} \gamma^t \cdot (v_{\pi'_1}(x_t) - q_{\pi'_1}(x_t, d_{1,t}, d_{2,t}) - q_{\pi'_1}(x'_t, d_{1,t}, d_{2,t}) + v_{\pi'_1}(x'_t)), \quad \text{(adding zero)}$$

$$= \sum_{t=0}^{\ell} \gamma^t \cdot ([v_{\pi'_1}(x_t) - q_{\pi'_1}(x_t, d_{1,t}, d_{2,t})] - [v_{\pi'_1}(x'_t) - q_{\pi'_1}(x'_t, d_{1,t}, d_{2,t})])$$

$$= \sum_{t=0}^{\ell} \gamma^t \cdot (v_{\pi'_1}(\cdot) - q_{\pi'_1}(\cdot, d_{1,t}, d_{2,t})) \cdot (x_t - x'_t).$$

Applying Hölder's inequality, using the definition of $\delta$, and the fact that $r$ is bounded:

$$v_1(b) - \min_{\pi_2' \in \Pi_2} v_{\pi_1', \pi_2'}(b) \leq \sum_{t=0}^{\ell} \gamma^t \cdot \|v_{\pi_1'}(\cdot) - q_{\pi_1'}(\cdot, d_{1,t}, d_{2,t})\|_{\infty} \cdot \|x_t - x_t'\|_1$$

$$\leq \delta \sum_{t=0}^{\ell} \gamma^t \cdot \|v_{\pi_1'}(\cdot) - q_{\pi_1'}(\cdot, d_{1,t}, d_{2,t})\|_{\infty}$$

$$\leq 2m\delta \sum_{t_0=0}^{\ell} \gamma^{t_0} \sum_{t_1=t_0}^{\ell} \gamma^{t_1-t_0}$$

$$= 2m\delta \sum_{t_0=0}^{\ell} \sum_{t_1=t_0}^{\ell} \gamma^{t_1}$$

$$= 2m\delta \sum_{t_0=0}^{\ell} \frac{\gamma^{t_0} - \gamma^{\ell+1}}{1-\gamma}$$

$$= \frac{2m\delta}{1-\gamma} \sum_{t_0=0}^{\ell} \gamma^{t_0} - \gamma^{\ell+1}$$

$$= \frac{2m\delta}{(1-\gamma)^2} [1 + (\ell+1)\gamma^{\ell+2} - (\ell+2)\gamma^{\ell+1}].$$

Combining both bounds gives the final exploitability guarantee and concludes the proof. $\qquad \square$

While it is theoretically sufficient to define a focal policy by computing values over the entire set of uninformed occupancy states, this approach is often impractical due to the exponential growth of that set with the horizon. To address this, we construct a worst-case trajectory through the uninformed occupancy space by solving a sequence of linear programs. At each step, the primal LP from Theorem 6 provides a greedy decision rule for the focal player. To complete the picture, we require a dual LP that identifies a worst-case response for the opponent, certifying the pessimism constraints that underpin the primal solution. This primal–dual pair induces a compact trajectory of uninformed occupancy states along which a focal policy can be explicitly extracted. The corollary below formalises the dual program that supports this construction.

**Corollary D.4.** *Let $o$ be an informed occupancy state. Then the pessimistic evaluation $q(o, \cdot)$, defined in Theorem 6, can equivalently be computed by solving the following linear program with:*

- $O(|\Phi_1| \cdot |\Phi_2^*| \cdot |\mathcal{H}_2(o)| \cdot |\mathcal{A}_2|)$ **variables**,
- $O(|\Phi_1| \cdot |\mathcal{H}_1(o)| \cdot |\mathcal{A}_1|)$ **constraints**,

*where $|\Phi_2^*| \doteq \max_{\Phi_2 \in \Phi_1} |\Phi_2|$. The dual linear program is:*

*Minimise* $\quad \sum_{h_2 \in \mathcal{H}_2(o)} \Pr(h_2 \mid o) \cdot u(h_2, \Phi_2)$

*Subject to* $\quad \sum_{\Phi_2 \in \Phi_1} \sum_{\phi_2 \in \Phi_2} \sum_{a_2 \in \mathcal{A}_2} \lambda(\Phi_2, \phi_2, h_2, a_2) = 1, \quad \forall h_2 \in \mathcal{H}_2(o),$

$u(h_2, \Phi_2) \geq \sum_{\phi_2 \in \Phi_2} \sum_{a_2 \in \mathcal{A}_2} \lambda(\Phi_2, \phi_2, h_2, a_2) \sum_{s \in \mathcal{S}} \phi_2(s, h_1, a_1, a_2) \cdot c_{2,(o,h_1)}(s, h_2),$

$\forall \Phi_2 \in \Phi_1, \ \forall h_1 \in \mathcal{H}_1(o), \forall a_1 \in \mathcal{A}_1.$

*The variable $\lambda(\Phi_2, \phi_2, h_2, a_2) \in [0, 1]$ represents the conditional probability of model–action pair $(\phi_2, a_2)$ under value model set $\Phi_2$ and private history $h_2$. The variable $u(h_2, \Phi_2)$ upper-bounds the expected value of Player 1's return under the induced model $\Phi_2$. The normalisation constraint ensures that for each $h_2$, the conditional distribution $\lambda(\cdot \mid h_2)$ is valid. This dual program reflects the adversary's strategy: choosing a worst-case model-action distribution per private history $h_2$ that maximises cost to Player 1 while respecting model uncertainty through $\Phi_2 \in \Phi_1$.*

*Proof.* The proof follows directly from the proof for the primal linear program, see Theorem 6. $\quad \square$

# E  Experiments

## E.1  Benchmarks

We evaluate our approach on several competitive benchmark problems, adapted from standard multi-agent settings. Their key characteristics are summarised in Table 2.

**Multi-Agent Recycling.** In the original cooperative version, two robots must clean a room represented as a grid by emptying garbage cans. Each robot has limited battery life and a restricted view of the environment, including limited observability of the other robot. Coordination is therefore required. We adapt the task to a zero-sum setting by altering the reward function: each robot now aims to clean more efficiently than the other.

**Multi-Agent Tiger.** The environment consists of two rooms—one containing a treasure and the other a tiger. Each agent stands before a door and may choose to either listen for cues or enter a room. Due to stochastic listening outcomes, agents receive noisy observations. Two competitive variants, *Adversarial Tiger* and *Competitive Tiger*, were introduced in Wiggers et al. [2016] to study adversarial behaviour under partial observability.

**Multi-Agent Broadcast Channel (MABC).** This benchmark captures a communication scenario where two agents (nodes) must broadcast messages over a shared channel. To prevent collisions, only one node may broadcast at any time. While the original version is cooperative—maximising joint throughput—we consider a competitive variant by modifying the reward structure.

**Matching Pennies.** Each player secretly chooses heads or tails. If the two choices match, Player 1 wins; otherwise, Player 2 wins. This is a simple, fully observable zero-sum game commonly used in theoretical analysis.

**Pursuit–Evasion.** This benchmark involves a grid-world where an evader attempts to escape a pursuer. Both agents can move in the four cardinal directions, and each perceives the opponent only when they occupy adjacent cells. The game continues after a capture, which rewards the pursuer and penalises the evader. We consider multiple grid sizes and obstacle settings to vary difficulty and observability.

Table 2: Benchmark characteristics. $|S|$: number of hidden states, $|A_i|$: number of actions for player $i$, $|Z_i|$: number of observations for player $i$, $R_{max}/R_{min}$: reward bounds, $\gamma$: discount factor.

| Problem | $|S|$ | $|A_1|$ | $|A_2|$ | $|Z_1|$ | $|Z_2|$ | $R_{max}$ | $R_{min}$ | $\gamma$ |
|---|---|---|---|---|---|---|---|---|
| Adversarial Tiger | 2 | 3 | 2 | 2 | 2 | 0.75 | -1.25 | 1 |
| Competitive Tiger | 2 | 4 | 4 | 3 | 3 | 0.66 | -0.66 | 1 |
| Recycling | 4 | 3 | 3 | 2 | 2 | 0.5 | -0.39 | 1 |
| MABC | 4 | 2 | 2 | 2 | 2 | 0.1 | 0.0 | 1 |
| Matching Pennies | 3 | 2 | 2 | 1 | 1 | 2.0 | -1.0 | 1 |
| Pursuit–Evasion $2 \times 2 \times 2$ | 16 | 4 | 4 | 6 | 6 | 0.0 | -100 | 1 |
| Pursuit–Evasion $3 \times 3 \times 1$ | 64 | 4 | 4 | 6 | 6 | 0.0 | -100 | 1 |
| Pursuit–Evasion $3 \times 3 \times 2$ | 81 | 4 | 4 | 6 | 6 | 0.0 | -100 | 1 |

## E.2 Additional Plots

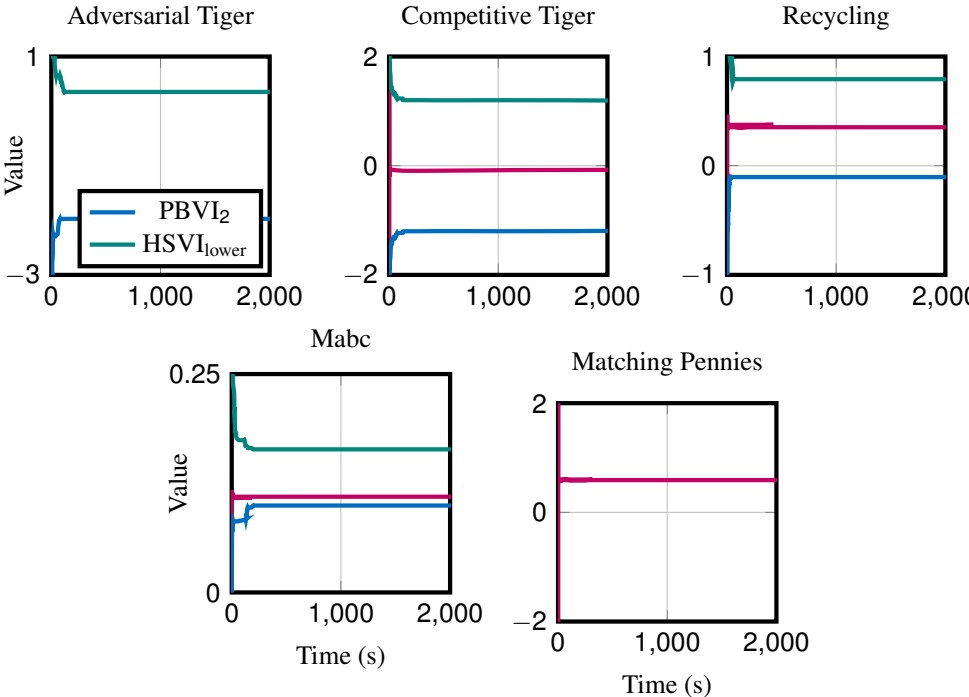

Figure 6: A visual representation of the performance of our best performing algorithm (PBVI$_3$) against the HSVI algorithm of Delage et al. [2023] for horizon $\ell = 4$ across five different games. **Best viewed in color.**

Table 3: Snapshot of empirical results. Games are ordered by increasing planning horizon $\ell$, and within each horizon by ascending number of local histories. For each setting, we report the value $v(b)$ and exploitability $\varepsilon$. OOT indicates a timeout (2-hour limit), OOM denotes out-of-memory runs, and '–' means the exploitability budget was exceeded. Best results are highlighted in **magenta**.

| Game ($\ell$) | PBVI$_1$ | | PBVI$_2$ | | PBVI$_3$ | | HSVI [Delage et al., 2023] | | CFR+ [Tammelin, 2014] | |
|---|---|---|---|---|---|---|---|---|---|---|
| | $v(b)$ | $\varepsilon$ | $v(b)$ | $\varepsilon$ | $v(b)$ | $\varepsilon$ | $v(b)$ | $\varepsilon$ | $v(b)$ | $\varepsilon$ |
| adversarial-tiger(2) | **-0.40** | **0.00** | **-0.40** | **0.00** | **-0.40** | **0.00** | **-0.40** | **0.00** | **-0.40** | **0.00** |
| adversarial-tiger(3) | **-0.56** | **0.00** | **-0.56** | **0.00** | **-0.56** | **0.00** | -0.56 | 1e-3 | **-0.56** | **0.00** |
| competitive-tiger(2) | **-0.02** | **0.00** | **-0.02** | **0.00** | **-0.02** | **0.00** | 0.00 | 0.00 | 0.00 | 0.00 |
| competitive-tiger(3) | -0.02 | **0.00** | -0.04 | **0.00** | **-0.03** | **0.00** | OOT | | **0.00** | **0.00** |
| recycling(2) | **0.26** | **0.00** | **0.26** | **0.00** | **0.26** | **0.00** | **0.26** | **0.00** | **0.26** | **0.00** |
| recycling(3) | **0.32** | **0.00** | **0.32** | 3e-2 | **0.32** | **0.00** | **0.32** | 1e-2 | **0.32** | 2e-2 |
| mabc(2) | **0.077** | **0.00** | **0.077** | **0.00** | **0.077** | **0.00** | **0.077** | **0.00** | **0.077** | **0.00** |
| mabc(3) | 0.095 | **0.00** | 0.094 | **0.00** | **0.096** | **0.00** | **0.096** | **0.00** | **0.096** | **0.00** |
| matching-pennies(2) | **0.20** | **0.00** | **0.20** | **0.00** | **0.20** | **0.00** | **0.20** | **0.00** | **0.20** | 1e-3 |
| matching-pennies(3) | **0.40** | 1e-3 | **0.40** | 1e-3 | 0.39 | **0.00** | **0.40** | **0.00** | **0.40** | **0.00** |

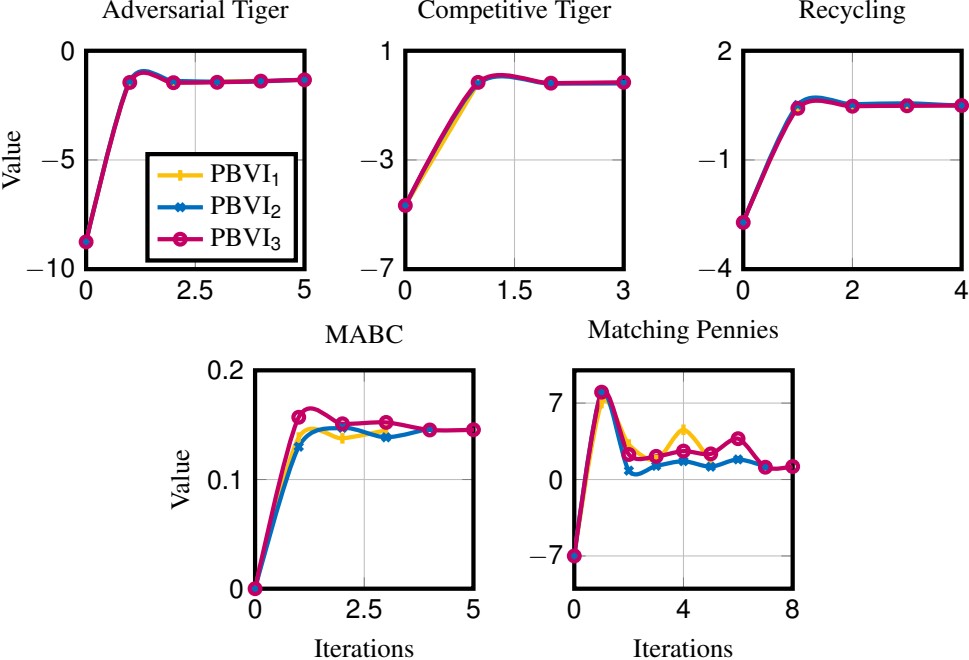

Figure 7: Performance over $\ell = 7$ for five benchmark problems. $PBVI_1$, $PBVI_2$, and $PBVI_3$ perform comparably on Adversarial Tiger, Competitive Tiger, and Recycling. $PBVI_2$ also matches $PBVI_1$ on MABC and Matching Pennies, while $PBVI_3$ struggles more on the latter. Pruning in $PBVI_2$ and $PBVI_3$ often enables continued improvement where $PBVI_1$ plateaus. **Best viewed in colour.**

