# OpenReview forum: "\(\varepsilon\)-Optimally Solving Two-Player Zero-Sum POSGs"
_NeurIPS.cc/2025/Conference — NeurIPS 2025 poster_

### Official Review · Reviewer_1Gi1 · 2025-06-29

**Clarity:** 3
**Significance:** 4
**Originality:** 4
**Rating:** 5
**Confidence:** 2

**Summary:**

This paper aims to solve two-player zero-sum partially observable stochastic games (zs-POSGs) up to $\varepsilon$-optimality. It introduces the first principled and lossless reduction from zs-POSGs to transition-independent zero-sum stochastic games (zs-SGs), marking the first such reduction to adversarial games. The core technical contribution is a novel hierarchy-of-planners framework that enables this reduction. This, in turn, allows the use of a broad class of dynamic programming-based methods, leading to both theoretical guarantees and scalable planning algorithms. The authors also adapt point-based value iteration methods and evaluate their approach across several benchmark domains, including matching pennies and pursuit-evasion, demonstrating performance that matches or surpasses existing state-of-the-art methods such as HSVI and CFR+.

**Questions:**

Are there any decentralized algorithms for zero-sum stochastic games that could potentially be applied to your reduced game? Do you think such an approach might improve the performance, particularly in terms of its dependence on the planning horizon?

**Ethical Concerns:**

["NO or VERY MINOR ethics concerns only"]

**Final Justification:**

The paper has novel aspects to it (see the Strengths section for more details). Moreover, the authors have responded to my questions in a satisfactory manner. Hence, I am retaining my current rating of '5: Accept'.

**Limitations:**

Yes

**Paper Formatting Concerns:**

No such concerns.

**Quality:**

4

**Strengths And Weaknesses:**

The main strength of this paper lies in the hierarchy-of-planners approach. The hierarchy comprises four planners:

* **Focal planner**: Knows only the focal policy and nothing else.
* **Uninformed planner**: Has access to the opponent's policy and the focal planner's data.
* **Informed planner**: Knows the public signals and the uninformed planner's data.
* **Marginal planner**: Has access to the opponent's history and the informed planner's data.

While only the focal planner is executed, the other planners provide important theoretical insights into the structure of the value functions. This hierarchy also reveals new structural properties of zs-POSGs, such as optimality equations, strategy selection rules, and the uniform continuity of value functions. The proposed reduction to transition-independent zs-SGs preserves key elements such as the value, equilibrium structure, and information constraints.

The hierarchy of planners also enables practical computation. The uniform continuity of the value function, established at the level of the marginal planner, allows for the use of point-based representations. This, in turn, makes it possible to apply point-based value iteration methods. A key component of the approach is the definition of action-value functions over the uninformed occupancy states, from which greedy decision rules are extracted via linear programming.

Although I did not identify any major weaknesses, I believe the presentation could be improved by including some of the diagrams from the supplementary material in the main body. Currently, the paper's main text feels somewhat dense due to the heavy concentration of technical content.

---

> ### Author Rebuttal · Authors · 2025-07-29
>
> We thank the reviewer for their thoughtful evaluation and are pleased that you found the planner hierarchy and theoretical contributions both original and impactful. We address your suggestions and questions below.
>
> **(1) Clarity and Use of Supplementary Diagrams.** We appreciate your point about the presentation. The main body is indeed dense due to the technical depth of the proposed reduction and planner hierarchy. In the final version, we plan to incorporate selected diagrams from the supplementary material to illustrate:
>
> - The structure and recursive flow of the planner hierarchy;
> - The decomposition of occupancy states;
> - Key transitions and value propagation within the planning process.
>
> We believe these visual aids will significantly improve clarity, especially for readers less familiar with zs-POSG structure.
>
> **(2) On Decentralised Algorithms and Horizon Scaling.** Thank you for this excellent question. While our reduction converts a zs-POSG into a transition-independent zs-SG with centralised occupancy states, the resulting planning procedure is decentralised in an important sense: we construct policies for the focal agent independently of the opponent’s strategy. This property arises from the structure of the reduced zs-SG and the design of our planner hierarchy. This opens up the possibility of distributed variants, where each agent plans over its own focal occupancy space, and coordination is achieved through shared guidance signals (e.g., by exchanging informative subspaces or value vectors). While still preliminary, this line of work shows promise, especially for distributed occupancy-set exploration or parallel policy refinement in structured settings. We will mention this as a speculative but compelling direction in the final version.
>
> Regarding horizon scalability, our method mitigates the exponential growth of the planning space through point-based approximations and pruning strategies. Nonetheless, we agree that future work could explore structured decompositions—such as factored occupancy representations or local Bellman updates—to improve scalability while maintaining solution fidelity.
>
> Finally, we note that decentralised learning or planning may be particularly viable in structured subclasses of zs-POSGs, such as those with sparse interaction or factored observation models. Combining our reduction with decentralised solvers in these settings could yield hybrid approaches that remain efficient without compromising equilibrium quality. While we have not yet pursued this empirically, structured multi-agent benchmarks could serve as a natural testbed. We will highlight this as a promising direction for future research.

---

> > ### Comment · Reviewer_1Gi1 · 2025-08-01
> >
> > Thank you for the rebuttal. I am satisfied with your response and will be retaining my current score.

---

### Official Review · Reviewer_smLN · 2025-07-02

**Clarity:** 2
**Significance:** 2
**Originality:** 3
**Rating:** 2
**Confidence:** 2

**Summary:**

This paper presents a method to simplify a zero-sum two player Stackelberg game by reformulating the primal construct of the game to allow for efficient representations and approximation of the players' value function. The authors provide the conditions and definitions for when this simplification is possible. Provide the methodology for this reduction, and theoretical guarantees regarding the algorithms performance, accompanied by empirical results.

**Questions:**

- Comparatively speaking, what is the more inefficient LP formualtion of the problem and what was the complexity reduction compared if someone were to brute-force the solution?
- Where is the source-code to reproduce these experiments?
- How come the CFR baseline is not presented in 3/4 plots?

**Ethical Concerns:**

["NO or VERY MINOR ethics concerns only"]

**Final Justification:**

First, I believe there was a significant violation of the supplemental appendix guidelines—as I noted in my original review. In its current form, the authors have effectively submitted two papers (a long and a short version), which risks reviewers defaulting to the longer document rather than the standard 8-page submission. While I acknowledge that the theory and proofs warrant extended discussion, this approach is unfair to other authors who adhere to the NeurIPS format of 8 pages for theoretical work. Supplemental material should supplement the main paper, not serve as a second, longer version. Even if this paper is accepted, this issue should be absolutely corrected.

Regarding the theoretical contributions, the work should strive to ensure clear practical relevance. As I mentioned to the authors, aside from Theorem 5.4, I found it difficult to fully appreciate the significance of the other supporting "Theorems." Some read more like definitions, while others seem better suited as supporting lemmas. A clearer organization of the theoretical presentation would really strengthen the paper.

Finally, the absence of a supporting codebase raises serious concerns about reproducibility. This is a major red flag for me. That said, if other reviewers find the theoretical arguments convincing on their own merits, I would defer to their judgment (I may have overlooked hidden ingenuity in the paper’s theoretical arguments).

**Limitations:**

- I don't see any major limitations of this work, my critique largely lies in the novelty of the methods. I do not believe the LP formulation is game-changing in terms of representing the Stackelberg ZS-POMDP efficiently. It provides a nice construct, good theory, but a more impactful research direction would be to look into kernel methods for non-lienar representations or dual-representations of the MIP/LP format, such that we can transform even non-linear problems into linear problems. Thus the technical challenge of the work is not to high, as I mentioned before, the von Neumann mini-max applies in this setting leading to simple reformulations, whereas in general sum games, a greater technical challenge exists.

**Paper Formatting Concerns:**

- In the table, many of the entries (e.g. pursuit-evasion-xxx) have 0.00 across the entire row, doesn't seem very necessary to communicate this. Also more significant digits should be employed. I'd recommend to present only the salient results on the table and leave the rest to the appendix.

- I also want to point out a critical item w.r.t. formatting. In the appendix, where the heart of the theory is, it seems like an entirely different and significantly longer version of the full paper was submitted. This is **not appropriate**, the NeuRIPS paper is 9 pages, and the proofs and supplementatal materials are a separate entity. The authors, can refer to the appendix, but should not deviate from the specifications by submitting a new paper altogether. **A major formatting change needs to happen here.**

**Quality:**

2

**Strengths And Weaknesses:**

Strengths:
- The authors use rigorous methods to convert a Stackelberg zs-POMDP into a more tractable representation. This simplification could have strong ramifications in multi-agent optimization and game theory.

Weaknesses:
- As a zero-sum game, the mini-max von Neumann theorem applies, it is not overly challenging to show that Theorem 3.4, and subsequently 4.1, holds. This is merely an extension from normal form to extensive form games, and the minimax theorem would still apply to sequential games.
- In the empirical experiments, there should be standard error measurements on the empirical estimates.
- This paper in general needs a bit more polish, the empirical results in Table 1 contains some unconvincing results (i.e. the new method poses no significant advantage compared to the baseline CFR.)
- Other Stackerlberg game solver baselines could also be employed here, such as KKT [1] reformulations of the problem, and gradient descent methods [2]. It was not considered.

[1] Allende, Gemayqzel Bouza and Georg Still (2013). “Solving bilevel programs with the KKTapproach”. In: Mathematical programming 138, pp. 309–332.
[2] Liu, Risheng et al. (2021). “Towards gradient-based bilevel optimization with non-convex followers
and beyond”. In: Advances in Neural Information Processing Systems 34, pp. 8662–8675.

---

> ### Author Rebuttal · Authors · 2025-07-29
>
> We thank the reviewer for the time and detailed feedback. We address each concern below and clarify key misunderstandings.
>
> **(1) Mischaracterisation of the setting as Stackelberg.** We respectfully clarify that our paper does not study Stackelberg games. Our focus is on two-player zero-sum partially observable stochastic games (zs-POSGs) with simultaneous moves. There is no notion of a committed leader policy or sequential leader–follower interaction. Consequently, suggestions such as KKT reformulations or bilevel gradient methods—tailored to Stackelberg-like settings—do not align with our formal setup. We suspect this mischaracterisation may have influenced the reviewer’s interpretation of the theoretical contribution and choice of baselines. We hope this clarification realigns the assessment with the actual scope of the paper.
>
> That said, we acknowledge that our framework performs planning for a single (focal) agent
> under worst-case assumptions, which may resemble leader optimisation in Stackelberg games. From this perspective, the focal agent plans over its own partial information while accounting for a best-responding opponent, which evokes the structure of zero-sum Stackelberg equilibria in partially observable domains. This resemblance may explain the reviewer’s terminology, though structurally our model remains a simultaneous-move game without any commitment semantics or sequential asymmetry.
>
> While Stackelberg POSGs strictly subsume zs-POSGs, our focus is not to generalise the setting but to preserve its semantics while enabling tractable and principled dynamic programming. We agree that the planning structure we introduce may inspire extensions to Stackelberg settings, but our analysis and algorithms are currently scoped to the simultaneous-move zs-POSG setting.
>
> **(2) On the novelty and theoretical contribution.**
> The reviewer suggests that our results follow from von Neumann’s minimax theorem. However, our goal is not to reprove minimax duality, but to construct the first lossless reduction from zs-POSGs to transition-independent zs-SGs. This reduction preserves value functions, equilibrium strategies, and dynamic programming structure, enabling a principled transfer of solution techniques. It resolves a long-standing gap between partially observable game theory and dynamic planning. The reduction is technically nontrivial and does not follow from existing minimax results.
>
> **(3) On the lack of error bars in experiments. We appreciate the observation.** While our approach is deterministic given a model, PBVI relies on sampled occupancy sets, introducing approximation variability. In line with standard practice in computational game theory, we report exploitability per instance. That said, we agree that standard deviations across runs could quantify this variability and will consider including such statistics in the final version.
>
> **(4) On the comparative advantage over CFR.** Our contribution targets scalability, not marginal performance gains. Table 1 shows that CFR fails due to time or memory limits in large-horizon domains, while our PBVI variants continue to produce bounded-ε solutions. These are settings where zs-POSGs become intractable for regret-based methods, yet our structural reduction enables dynamic programming. This is a qualitative advantage arising from the reduced model’s structure.
>
> **(5) On the choice of baselines.** The reviewer suggests bilevel optimisation methods such as:
>
> - Allende & Still (2013): Solving bilevel programs with the KKT approach
> - Liu et al. (2021): Gradient-based bilevel optimisation with non-convex followers
>
> However, these works address static bilevel problems with leader–follower structure and do not incorporate stochastic transitions, partial observability, or multi-stage dynamics. For example, Allende & Still reformulate bilevel programs via KKT into MPCCs, while Liu et al. address differentiable bilevel problems from meta-learning. In contrast, our setting involves planning in multi-stage simultaneous-move zs-POSGs using a structural reduction to enable dynamic programming. These approaches operate in a different algorithmic and representational regime.
>
> **(6) On code availability.** As indicated in Checklist Item 5, we commit to releasing all code and data necessary to reproduce the results upon acceptance, in accordance with NeurIPS reproducibility policy.
>
> **(7) On supplementary material formatting.** Our submission follows NeurIPS 2025 formatting guidelines. To aid clarity, we included a supplementary version where hidden content (e.g., extended proofs and diagrams) has been revealed. This version introduces no new claims and serves to enhance transparency, not to circumvent formatting rules. We are happy to restructure this material if the reviewer prefers a different format.

---

> > ### Comment · Reviewer_smLN · 2025-08-06
> >
> > I thank the authors for clarifying my misunderstandings regarding the characterization of the Stackelberg game setting. I largely agree with their counterargument and acknowledge that the problem the authors present is richer than the static bilevel optimization problems I referenced.
> >
> > That said, I still find it somewhat challenging to fully grasp the novelty of the paper and the significance of the theorems. A concise summary of each theorem’s role and how they connect would greatly help in appreciating the contributions. For instance, Theorem 4.3 appears more like a definition—clarifying its purpose (e.g., whether it is an end result or supports another result, in which case it might be better framed as a supporting lemma) would strengthen its presentation.
> >
> > Regarding Theorem 5.2, which establishes that the POMDP can be solved via a linear program, it would be valuable to know whether this linear program is applied in the experiments - reprodcibility is hard to establish when no code is supplied in the supplemental. Finally, while Theorem 5.4’s results seem impactful, I am curious whether there are any additional insights on sample complexity or regret bounds - this would strengthen the paper further.
> >
> > Overall, these clarifications would greatly improve the quality of the work in its current form - I will update my review accordingly. I thank the authors again for their diligent response.

---

### Official Review · Reviewer_WKef · 2025-07-02

**Clarity:** 3
**Significance:** 3
**Originality:** 3
**Rating:** 5
**Confidence:** 3

**Summary:**

The authors tackle the problem of computing approximate equilibria to 2p0s POSGs, and present a novel framework that is lossless, in the sense that it preserves value functions, equilibria, and information. Based on this reduction, they apply PBVI algorithms, to compute $\varepsilon$ optimal equilibria in a variety of games.

**Questions:**

Is there a reason why in the experiments why you don’t compare to the annealing approach of Sokota et al.?

**Ethical Concerns:**

["NO or VERY MINOR ethics concerns only"]

**Final Justification:**

My questions above have all been addressed. The authors said that they will address the formatting concerns in the final revision. My question regarding the comparisons to Sokota et al. have also been addressed.

**Limitations:**

Yes, the authors have addressed limitations.

**Quality:**

3

**Strengths And Weaknesses:**

The paper is well-written and tackles an important problem of equilibrium computation in 2p0s POSGs. Their lossless framework allows application of techniques in “vanilla” Markov games to the 2p0s POSGs (PBVI algorithms are the only ones that can be applied).
In terms of weaknesses, given my background (or lack thereof), I am not well equipped to assess originality beyond the presentation of the authors. I find the math typesetting a bit odd and in some parts hard to read. It seems that many different typesetting styles are being combined, and personally, that, combined with the color subscripts, makes it harder to read rather than easier because it is distracting. The authors should consider shortening exposition in certain portions to make the graphs and tables in the experiment section larger. While the trends are clear in the graphs, and the figures in the table can be read, they are both quite small, the table especially.

---

> ### Author Rebuttal · Authors · 2025-07-29
>
> We thank the reviewer for the positive and constructive feedback. We appreciate your recognition of the importance of solving two-player zero-sum POSGs via a lossless reduction, and for highlighting the clarity of the experimental trends. Below, we respond to your suggestions point by point.
>
> **(1) Math typesetting and notational styles.** Thank you for this valuable observation. We acknowledge that the use of colour-coded subscripts (e.g., for player indices) and a mixture of font styles may appear visually non-standard. Our intention was to highlight agent roles and variable types clearly, but we understand that this can be visually distracting. Based on your feedback, we will streamline the notation in the final version—removing coloured subscripts, avoiding non-standard font changes, and adopting more conventional mathematical notation for improved consistency and readability.
>
> **(2) Graph and table sizing.** We appreciate your point regarding the sizing of figures and tables, especially in the experimental section. While we aimed to balance visual elements with structural exposition, we agree that certain graphical components—notably Table 1—are densely presented. In the final version, we will revise the layout to allocate more space to tables and figures, including enlarging key plots. To do so, we will condense surrounding text where possible, without compromising clarity or completeness.
>
> **(3) On the absence of Sokota et al. (ICML 2023) in our experimental comparisons.**
> Thank you for raising this important point. We are familiar with Sokota et al. (2023), who propose a KL-regularised formulation of two-player zero-sum games. Their regularisation induces a perfect-information tree structure that facilitates dynamic programming. However, their approach fundamentally modifies the original game’s equilibrium structure through entropy regularisation, which alters the equilibrium value and policies. In contrast, our framework performs a lossless reduction that strictly preserves the original value functions, equilibrium strategies, and information constraints of the zs-POSG.
>
> This key distinction implies that the two approaches are not directly comparable in terms of solution quality with respect to the original game (e.g., measured by exploitability). Moreover, the scalability of Sokota et al.’s method to large, unstructured zs-POSGs remains to be demonstrated, as their experiments are limited to small gridworlds and simplified poker subgames with fixed trees. That said, we view their approach as complementary to ours—particularly for structured domains where the induced perfect-information tree is compact. We will clarify this comparison in the related work section.

---

> > ### Comment · Reviewer_WKef · 2025-08-05
> >
> > Thank you for addressing my concerns. I continue my support of the paper, and will maintain my score.

---

### Official Review · Reviewer_e8ES · 2025-07-03

**Clarity:** 3
**Significance:** 3
**Originality:** 4
**Rating:** 5
**Confidence:** 3

**Summary:**

In this paper, the authors present a framework that introduces the first lossless reduction from zs-POSGs to transition-independent zs-SGs, enabling the principled application of a wide range of dynamic programming (DP) methods.

The paper begins by introducing the problem and detailing the limitations of existing reductions. The authors then reformulate zs-POSGs as transition-independent zs-SGs by decentralizing planning into per-player occupancy sets, thereby preserving value functions, equilibria, and information constraints.

Next, they show how transition-independent zs-SGs can be solved via a hierarchy of planners, which enables more efficient value function decomposition and planning.

Finally, the authors demonstrate how Point-Based Value Iteration (PBVI) can be applied over the reduced zs-SG and validate their theoretical results empirically on a suite of challenging benchmark domains.

**Questions:**

1. I am curious whether you have considered comparing your approach to recent deep reinforcement learning (deep RL) based methods for imperfect-information games, such as Neural Fictitious Self-Play or other policy optimization frameworks (e.g., actor-critic variants or regret minimization with function approximation). While I understand that these methods often lack the same theoretical guarantees and lossless reductions, they have shown practical success on some large-scale settings. Do you see your framework as complementary to these approaches, or would you expect it to outperform them in terms of exploitability or sample efficiency? It would be helpful to hear your perspective on how your method compares empirically or conceptually to deep RL-based baselines, and whether you envision integrating neural approximators into your framework in the future.

**Ethical Concerns:**

["NO or VERY MINOR ethics concerns only"]

**Final Justification:**

I appreciate the technical contribution of this paper. My main concern lies in the presentation of the results, which I believe can be improved in the camera-ready version. Therefore, I will maintain my current score.

**Limitations:**

yes

**Paper Formatting Concerns:**

I didn't notice any major formatting issues.

**Quality:**

4

**Strengths And Weaknesses:**

Strengths:
1. The problem is very well motivated. I appreciate that the authors clearly articulate the desirable criterion of losslessness early on, which helps frame the technical goals precisely and underscores why prior approaches fell short.
2. Although I am not deeply familiar with all related literature, the contributions appear to be technically solid and substantial. Sections 3, 4, and 5 each introduce significant and novel technical elements, including the construction of the reduction, the hierarchy of planners, and the uniform continuity results that underpin the approximation guarantees.
3. Beyond the strong theoretical results, the authors thoughtfully address practical considerations around scalability. The proposed pruning strategies for point-based value iteration are well-motivated, and the experimental evaluation is convincing, showing consistent improvements in exploitability and runtime over prior baselines.

Weakness:
1. While the theoretical development is elegant, the presentation is quite dense. I think additional figures, diagrams, or illustrative examples would help make the key constructions (particularly the planner hierarchy and the occupancy set definitions) more accessible to readers who are less familiar with the technical background.
2. One potential concern is memory usage. Because the approach relies on sampling and storing sets of marginal and informed occupancy states, I wonder whether the memory footprint grows significantly with the horizon and problem size, perhaps more so than some regret-minimization approaches. It would be helpful if the authors could clarify this point or provide comparative memory usage statistics.

---

> ### Author Rebuttal · Authors · 2025-07-29
>
> We thank the reviewer for the thoughtful and encouraging evaluation. We are pleased that you found the contributions technically solid, well-motivated, and practically relevant. We address below the suggestions and questions raised.
>
> **(1) Clarity of Presentation – Additional Figures and Examples.** We appreciate your comment regarding the density of Sections 3–5. These sections indeed introduce several structural components, such as the planner hierarchy and occupancy set decompositions, which are central to the reduction and subsequent planning algorithm. In the camera-ready version, we plan to include additional figures and illustrative examples that visually convey:
>
> - The structure and recursive flow of the planner hierarchy
> - The distinction between informed, marginal, and focal occupancy sets
> - Concrete step-by-step transformations from zs-POSG instances into the reduced zs-SG formulation
>
> We believe these additions will significantly improve accessibility without altering the technical
> depth.
>
> **(2) Memory Usage and Scalability.** You are correct that PBVI-based methods rely on sampling and storing occupancy states, including marginal and informed sets. Our pruning strategies, as described in Section 6, aim precisely to mitigate this challenge by discarding dominated value vectors and uninformative occupancy states. Also, binary encodings of state–history tuples and history compression can help improve memory usage and scalability (see Dibangoye et al., JAIR, 2016).
>
> **(3) Comparison to Deep RL Methods and Neural Function Approximation.** This is a very insightful question. We view our framework as complementary to deep RL-based approaches, such as Neural Fictitious Self-Play (NFSP), actor-critic algorithms, and deep regret minimisation techniques. While these methods have achieved success in large-scale games (e.g., poker), they generally lack formal guarantees regarding value preservation or solution optimality under partial observability. Our goal is to establish a principled foundation for planning in zs-POSGs, enabling algorithmic transfer from dynamic programming while retaining lossless structure. That said, we agree that integrating neural approximators within our planner hierarchy (e.g., to generalise over large or continuous occupancy spaces) is a promising future direction. Preliminary experiments in this direction suggest that hybrid neural–symbolic versions of our approach could combine the best of both paradigms: structural interpretability and scalability through function approximation. We will mention this explicitly in the discussion section.

---

> > ### Comment · Reviewer_e8ES · 2025-08-07
> >
> > Thank you for the rebuttal. I found your response satisfactory and will be maintaining my current score.

---

### Decision · Program_Chairs · 2025-09-17

**Decision:**

Accept (poster)

**Comment:**

This paper presents the first lossless reduction from two-player zero-sum partially observable stochastic games (zs-POSGs) to transition-independent zs-SGs via a hierarchy-of-planners framework, enabling dynamic programming methods such as PBVI to compute approximate equilibria. Experiments on benchmark domains show performance on par with or better than state-of-the-art methods.

From a technical side I think this is a strong paper that solves an important problem via a novel approach (as 3 out of the 4 reviewers agree), but I also agree with most reviewers that the presentation needs some improvement. Reviewer smLN's main reasons for rejection are 1) "violation" of the appendix format, 2) poor theoretical presentation, and 3) absence of a supporting codebase. As far as I know, 1) and 3) do not really violate any policies. Therefore, I still recommend accept, with the expectation that the authors will improve the presentation and eventually provide the codes for reproducing the experiments as promised.